# Water-organizing motif continuity is critical for potent ice nucleation protein activity

Jordan Forbes[1], Akalabya Bissoyi[2,6], Lukas Eickhoff [3,6], Naama Reicher[4,6], Thomas Hansen [1], Christopher G. Bon [1], Virginia K. Walker [5], Thomas Koop [3], Yinon Rudich [4], Ido Braslavsky [2] & Peter L. Davies[1] ✉

Bacterial ice nucleation proteins (INPs) can cause frost damage to plants by nucleating ice formation at high sub-zero temperatures. Modeling of *Pseudomonas borealis* INP by AlphaFold suggests that the central domain of 65 tandem sixteen-residue repeats forms a beta-solenoid with arrays of outward-pointing threonines and tyrosines, which may organize water molecules into an ice-like pattern. Here we report that mutating some of these residues in a central segment of *P. borealis* INP, expressed in *Escherichia coli*, decreases ice nucleation activity more than the section's deletion. Insertion of a bulky domain has the same effect, indicating that the continuity of the water-organizing repeats is critical for optimal activity. The ~10 C-terminal coils differ from the other 55 coils in being more basic and lacking water-organizing motifs; deletion of this region eliminates INP activity. We show through sequence modifications how arrays of conserved motifs form the large ice-nucleating surface required for potency.

Ice nucleation proteins (INPs) induce heterogeneous ice formation in supercooled water. They have been described in certain Gram-negative bacterial isolates, including those of *Pseudomonas syringae*[1], *P. fluorescens*[2], *Erwinia herbicola*[3], and *P. borealis*[4]. Due to the broad distribution of ice nucleation-active bacteria on the surfaces of various plant species[5] and their ability to initiate ice formation at temperatures as warm as −2 °C, INPs represent the primary source of frost damage in nature[6]. Once nucleated on frost-sensitive plants, ice grows both inter- and intracellularly, causing physical disruption of cell membranes[7]. Since increased INP expression can be triggered by nutrient starvation[8], phytopathogenic strains of ice-nucleating bacteria may have evolved INPs to aid in plant tissue colonization and nutrient access[9]. Airborne INP-bearing bacteria may also have an important role in the hydrological cycle by initiating precipitation in clouds[10,11]. Despite the ubiquity of ice-nucleating bacteria and their importance in frost damage to crops and in precipitation, relatively little is known about the structure of INPs and the mechanism through which they trigger ice formation.

The large size (~ 130,000 Da) and the binding of INPs to the outer bacterial membrane may have hindered their structural characterization. However, several models have been proposed for the central repetitive region comprised of ~50–80 tandem copies of a well-conserved 16-residue sequence[12–14]. In 1993, Kajava and Lindow modeled the central domain of InaZ from *P. syringae* as a series of anti-parallel β-sheet clusters in a planar zigzag arrangement that enabled interdigitation of monomers to form large aggregates[14]. Two later models were inspired by similarities in repeat sequence to the beta-solenoid fold[15] used by some antifreeze proteins (AFPs) such as the spruce budworm and mealworm beetle AFPs[16–18]. The *P. syringae* INP model of Graether and Jia[13] was built on a coil of UDP-acetylglucosamine acyltransferase (PDB 1LXA), shortened from 18 to 16 residues and repeated three times to model a 48-residue segment as a beta-solenoid. In contrast, the model of ref. 12 was guided by the beta-roll structure common to RTX proteins[19], but it also produced a beta-solenoid structure for a section of eight 16-residue coils from *P.*

[1]Department of Biomedical and Molecular Sciences, Queen's University, K7L 3N6 Kingston, ON, Canada. [2]The Robert H. Smith Faculty of Agriculture, Food and Environment, Institute of Biochemistry, Food Science, and Nutrition, The Hebrew University of Jerusalem, Rehovot 7610001, Israel. [3]Bielefeld University, Faculty of Chemistry, D-33615 Bielefeld, Germany. [4]Department of Earth and Planetary Sciences, The Weizmann Institute of Science, Rehovot 7610001, Israel. [5]Department of Biology, Queen's University, Kingston, ON, Canada. [6]These authors contributed equally: Akalabya Bissoyi, Lukas Eickhoff, Naama Reicher. ✉e-mail: peter.davies@queensu.ca

*borealis* INP (*Pb*INP). The repeating outward-projecting tyrosine and neighboring serine residues were proposed to form a dimerization surface in this latter model.

An exciting discovery of the solenoid models was the regular placement of a TxT motif where the two threonine residues on the beta-strand project outwards as they do in several insect AFPs[16–18]. In these AFPs, the threonyl side chains form two parallel arrays that constitute the ice-binding site, as demonstrated by site-directed mutagenesis[20]. The spacing of these threonine residues on a flat beta-sheet matches closely the spacing of oxygen atoms in the ice lattice on the basal and primary prism planes[17]. One proposal for the mechanism by which AFPs bind to ice is that these threonine residues organize water molecules in an ice-like arrangement so that they merge with the quasi-liquid layer on the ice lattice and then, as the system cools, serve to freeze the AFPs onto ice[21,22]. Ice growth is then inhibited by the Kelvin effect[23]. However, this hypothesis was not supported by vibrational sum-frequency generation spectroscopy[24] or by a modeling study[25], which suggested that water ordering is not essential for AFP activity. For the much larger INPs that are thought to self-associate on the outer bacterial membrane[26], it has been suggested that the organization of many ice-like water molecules on one surface may trigger ice nucleation[27–30]. Thus, there may be a common theme to the mechanisms of action of AFPs and INPs: the ordering of ice-like water molecules by a regular array of residues on a flat protein surface. AFPs would only need a small number of ice-like water molecules to adsorb to ice and stop its growth by the Kelvin effect[31]. These small proteins can only slightly raise the temperature for the nucleation of new ice crystals[32]. However, large INPs and their aggregates order enough ice-like water molecules to initiate nucleation at high sub-zero temperatures[27–29,33]. The importance of the organized water molecules in INP activity has even been questioned[34]. Notably, these authors, using sum-frequency generation spectroscopy, found that both active and heat-inactivated INPs showed no difference in the ordering of interfacial water molecules upon cooling. However, another study using sum-frequency generation and 2D infrared spectroscopy did provide evidence for a structural ordering of water by INPs[30].

Clearly, more experiments are needed to clarify the INP mechanism of action. Here we tested the water ordering hypothesis by examining the impact of *Pb*INP truncations, mutations, and central domain interruptions on ice nucleation temperatures. We show that *E.*

*coli* expressing recombinant *Pb*INP with fewer repeats containing putative water-organizing residues reproducibly freeze at lower temperatures, where a minimum central domain length of 13–15 repeats is required for activity. Conservative mutations of threonine and tyrosine in the water-organizing motifs reduce freezing temperatures in both full-length and shortened *Pb*INP central domains. A similar decrease in ice nucleation efficiency from the insertion of a functional bulky domain into the central domain indicates the importance of an uninterrupted water-organizing surface. Based on the *Pb*INP AlphaFold model and deletions from different regions of the central domain, we propose that the -10 repeats adjacent to the C-terminal domain contribute to the formation of INP aggregates required for efficient ice nucleation.

## Results

### Modeling with AlphaFold predicts the domain structure of *Pb*INP

According to AlphaFold, the N-terminal domain of *Pb*INP begins with a 110-residue globular domain followed by a long flexible linker (Fig. 1a). Based on the alignment of ten N-terminal domain sequences of bacterial INPs, the globular domain sequence is highly conserved, while the linker region is variable in length and sequence (Supplementary Fig. 1). Next is the central repetitive domain modeled as a continuous beta-solenoid in which each 16-residue repeat forms one coil of the solenoid. This structure is similar to previous homology-based solenoid models made for short stretches of the central repetitive domain[12,13] but differs slightly in cross-section (Fig. 1b). The S at position 5 faces inward in the AlphaFold model to brace that side of the solenoid through an internal serine ladder, equivalent to S13 on the opposite side. This results in a flatter solenoid where the opposing beta-sheets are parallel. In the Garnham model, S5 points outward to form a putative dimerization motif along with Y3 and gives the solenoid a slightly triangular cross-section[12] (Fig. 1c). Over the length of the 65 repeats, the AlphaFold model suggests a slight left-handed twist to the solenoid, amounting to a 180° rotation from end to end. Finally, the 41 residues of the C-terminal domain adopt a beta-strand structure that appears to serve as a cap over the end of the central solenoid (Fig. 1a). This domain is in intimate contact with the last coil of the solenoid but does not interact with neighboring coils. Based on multiple sequence alignments of the INPs used in Supplementary Fig. 1, the C-terminal domain shows extensive sequence identity, as do their

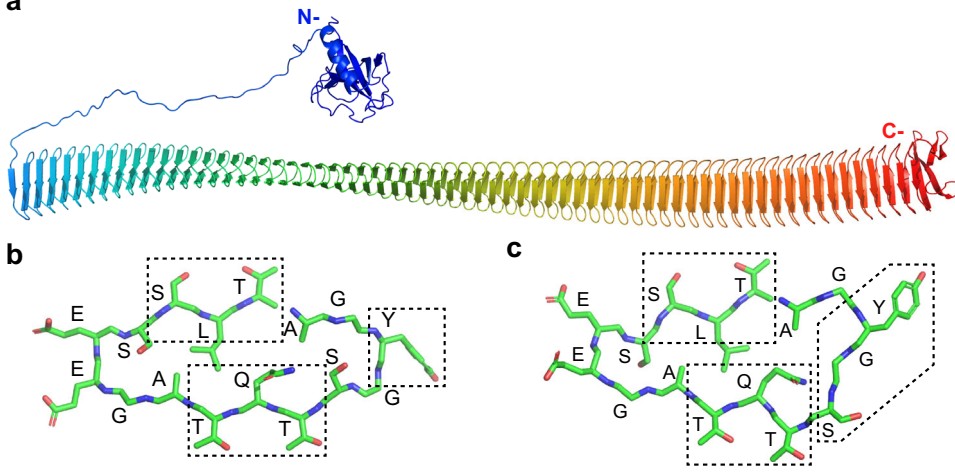

**Fig. 1 | AlphaFold model of *Pb*INP and cross-section through the solenoid. a** The AlphaFold model of *Pb*INP is displayed in chainbow coloration with arrows representing beta strands and coils for alpha helices. **b** Cross-section through repeat six of the central domain in the AlphaFold model. Residues are identified by their one-letter code. Boxes indicate the location of the TxT motif, the SLT motif, and the conserved Y ladder. Side chain oxygen atoms are red, nitrogen atoms are blue, and carbon atoms are green. **c** The equivalent cross-section in the Garnham model[12] that corresponds to the same 16-residue repeat with the same motifs indicated, but with the outward-facing S grouped with the Y ladder.

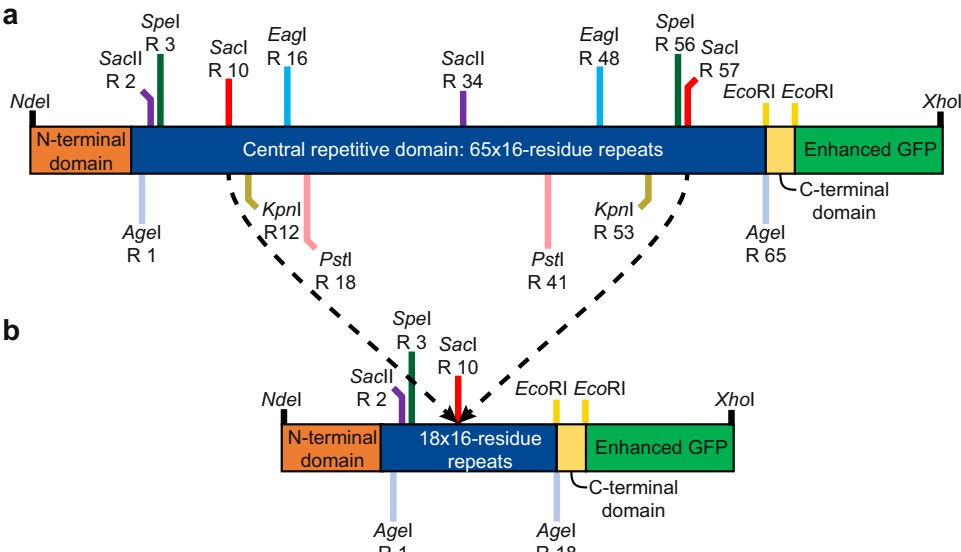

**Fig. 2 | Map of synthetic, codon-optimized sequence used to produce recombinant *Pb*INP deletion constructs. a** The positions of unique pairs of color-matched restriction enzyme cut sites are indicated, as well as the repeat number ("R") in which they are situated. Pairs of cut sites within the blue central domain encoding 65 16-residue repeats are separated by multiples of 48 bp, which maintain the sequence integrity of the 16-residue repeats. The N- and C-terminal domains are shown in orange and yellow, respectively, and green fluorescent protein (GFP) is in green. **b** An example of a deletion construct produced using this synthesized sequence treated with the restriction enzyme *Sac*I (recognition sites shown as red vertical lines), where 18 of the 65 16-residue repeats remain in the central domain and repeats R10 to R57 are deleted.

central repetitive domains despite differences in the total number of 16-residue repeats that range from ~50 to ~80. The portion of the alignment shown in Supplementary Fig. 1 is the N-terminal region and illustrates good conservation of the first half that corresponds to the globular region in the AlphaFold structure prediction (Fig. 1a) but poor conservation of the disordered linker region.

## Synthetic *Pb*INP gene facilitates deletion and mutagenesis studies

To enable systematic and extensive deletion analyses, with the option to mutate large sections of the INP gene, we synthesized a codon-optimized *Pb*INP gene containing silent mutations that introduced many pairs of restriction sites at specific matching locations within the 48-bp repeats that code for the putative solenoid coils (Fig. 2a). For example, digestion with *Sac*I followed by ligation removed 47 of the 65 central repeats to produce a construct with only 18 repeats remaining (Fig. 2b). With the synthetic INP gene C-terminally linked in frame with the DNA sequence for green fluorescent protein (GFP), all constructs produced in *E. coli* showed green fluorescence thereby confirming that any loss of ice nucleation activity was not due to failure in construct expression or a frame-shift mutation.

## Positive correlation between central domain length and *Pb*INP activity

Previous studies on the *P. syringae* INP gene *inaZ* (encoding 61 solenoid coils) expressed in *E. coli* have shown that deletions of 1–34 repeats from the central repetitive region reduced the ice nucleation temperature by up to 8.5 °C, in a manner generally consistent with deletion size[27,35]. These analyses were limited because the apparatuses used could only measure droplet freezing at temperatures far above the homogeneous ice nucleation temperature. Here, we have been able to measure the lower limits of ice nucleation activity of recombinant *E. coli* cultures bearing *Pb*INP constructs by using two small volume freezing assays described in Methods: WISDOM, wherein monodisperse nanoliter-sized droplets are suspended in an oil mixture, and BINARY, involving microliter-sized droplets sealed in individual compartments. Ice nucleation for the various cultures was compared by determining the fraction of frozen droplets ($f_{ice}$) as a function of temperature in both systems (Fig. 3a, b). Since droplet freezing occurred over a narrow temperature range of one to two degrees Celsius for active samples, the nucleation curves were readily compared between constructs and nucleation apparatuses. For example, the 29-repeat *Pb*INP variant showed nucleation at approximately −10 °C on both the WISDOM and BINARY systems, while the full-length control cultures nucleated freezing near −8 °C. Comparisons were further facilitated by plotting $T_{50}$ values (the temperature at which 50% of the droplets are frozen) for each repeat deletion culture against the number of repeats ($n_{rep}$) remaining in its central domain (Fig. 3c). There was good agreement between nucleation curves obtained in the two devices for every deletion construct. Neither apparatus showed more than a few °C difference in ice nucleation temperature between cultures with full-length *Pb*INP and deletions that removed up to 60% of the central repeats. In contrast, assays using cultures with fewer than 24 INP repeats showed a sharp but continuous reduction in activity. The ice nucleation temperature for the 18-repeat construct was ~−17 °C, and this fell to ~−23 °C for the 15-repeat construct. When only one or 12 repeats remained, the ice-nucleating ability was lost and closely matched the negative controls with either of the small volume methods.

## Critical number of INP repeats for ice nucleation activity

As indicated in Fig. 3 and Table 1, the six-repeat difference between 18-repeat and 12-repeat constructs caused a substantial decline in ice nucleation activity: by 20 °C towards homogeneous freezing in the WISDOM apparatus and by 11 °C towards the background value of ice formation in pure water on the BINARY apparatus. To assess this loss in more detail, constructs were produced with 15, 16, and 17 repeats using a synthetic gene fragment with pairs of restriction sites separated by multiples of 48 bp (Supplementary Table 1). Assays with both apparatuses showed a stepwise increase in $T_{50}$ values as the central domain was lengthened from 15 to 18 repeats, where incremental additions of one repeat were associated with a ~2 °C increase in $T_{50}$ (Table 1). These data suggest that the minimum number of repeats required for discernible *Pb*INP function would be 15, 14, or 13.

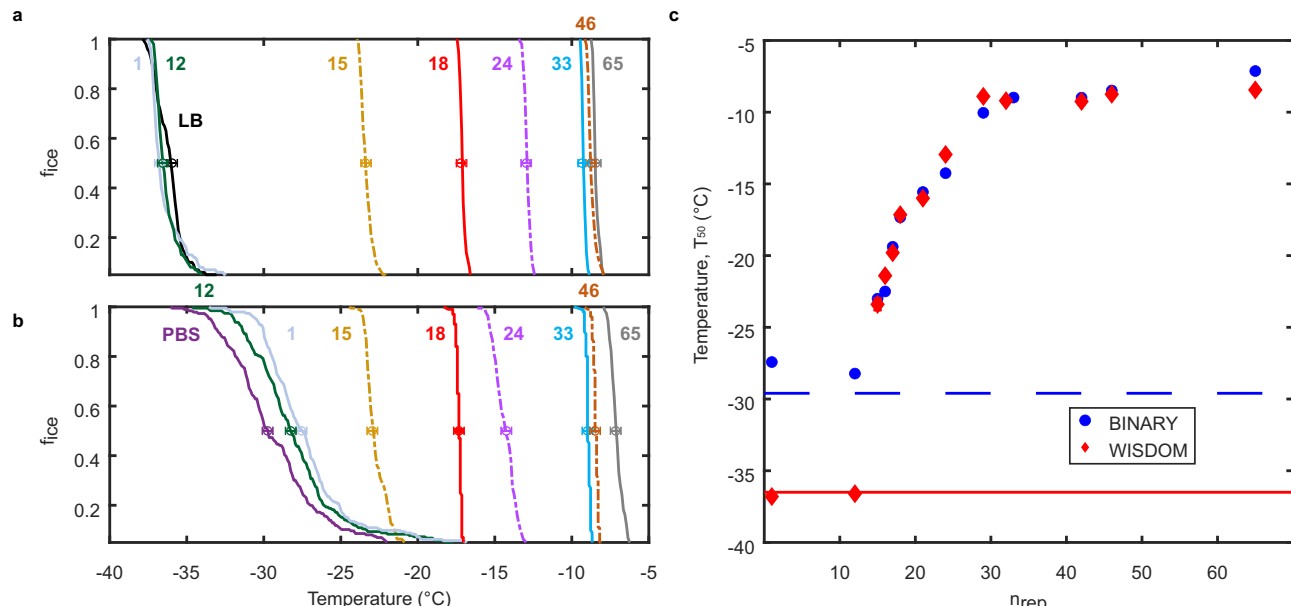

**Fig. 3 | Ice nucleation assay data from *Pb*INP repeat deletions expressed in *E. coli*. a, b** Fraction of frozen droplets ($f_{ice}$) vs. temperature from the WISDOM and BINARY assays, respectively. The values alongside or above the colored plots refer to the number of central domain repeats remaining in that construct. Each sample's data points are depicted identically in parts A and B; gold dashed line for 15 repeats, red solid line for 18 repeats, etc. **c** The relationship between $T_{50}$ values and the number of INP repeats in the central domain. Data from BINARY and WISDOM are presented in blue-filled circles and red diamonds, respectively; the background freezing temperature of pure LB media/buffer samples for each apparatus is indicated by a horizontal line.

## Serial dilutions of INP-expressing bacteria and their effect on activity

Previously, it has been reported that wild-type *P. syringae* contain sub-populations of cells with a range of ice nucleation temperatures[1,36–38], and thus we investigated sample homogeneity in *P. borealis* and a recombinant *E. coli* culture expressing *Pb*INP. In droplets containing hundreds or thousands of cells, any heterogeneity in INP activity would be masked because the nucleation temperature reflects that of the most potent ice nucleator in the sample. Therefore, cultures were diluted so that the assay droplets were calculated to contain zero, one, two, or at the most a few bacterial cells. The analysis of an *E. coli* culture expressing *Pb*INP with 33 repeats exhibited a dichotomy where some droplets showed established nucleation temperatures corresponding to the genotype of the culture (compare Fig. 3), while other droplets from the same culture failed to nucleate until the homogeneous ice nucleation temperature was reached (Fig. 4a). This "all-or-nothing" response enabled calculation of the percentage of droplets without an ice nucleator as well as the number of ice nucleators per drop. For example, in Fig. 4b, at dilution $10^{-3}$, 60% of the drops freeze only at the homogeneous ice nucleation temperature, and 40% of the drops

freeze heterogeneously. From Poisson statistics, it can be shown that most of the latter drops (76%), i.e., those that froze heterogeneously, contain only one nucleator (Supplementary Note 1). The finding that the nucleation temperature of one nucleator in a drop equals the nucleation temperature of a few hundred nucleators in the same volume suggests sample uniformity, where nucleation is controlled by the length of the INP rather than the amount of product in the drop. Interestingly, a small percentage (-8% of the drops in the $10^{-2}$ dilution, red line in Fig. 4b) of the wild-type *P. borealis* culture induced freezing several degrees above the average value, suggesting that 0.5 to 1% of these bacteria had more potent ice nucleation activity than the rest of the culture (Fig. 4b). Data from the $10^{-2}$ and $10^{-3}$ dilutions in Fig. 4a, and from the $10^{-2}$, $10^{-3}$, and $10^{-4}$ dilutions in Fig. 4b, were replotted to show the cumulative number of ice nucleators per bacterial mass (Nm) as a function of temperature (Supplementary Fig. 2). These plots also show that all the *E. coli* serve as ice nucleators within a narrow range of activity and that there is a sharp transition to a lack of heterogeneous ice nucleation when zero bacteria are present in a droplet. For *P. borealis*, when the number of bacteria per droplet rises from 1 to 1000, we see rare ice nucleation events that shift from −10 °C (at $10^{12}$ Nm g$^{-1}$) to −6 °C (at $10^{9}$ Nm g$^{-1}$).

## C-terminal domain and nearby tandem repeats are critical for activity

One of our initial assumptions was that all INP repeats would be roughly equivalent in their contribution to heterogeneous ice nucleation. We investigated this supposition by examining the match of each 16-residue repeat to the consensus sequence AGYGSTQTAGEESSLT (Fig. 5a). Extension of consensus matching on either side of the central repetitive domain demonstrated that the number of repeats (65) and domain boundaries were accurate. At the N-terminal end, the match fell from eight residues out of 16 in the first repeat to just two residues within the 16 adjacent positions in the N-terminal domain (regions shown as blue and orange, respectively, in Fig. 5a). At the C-terminal end, the match fell from 8 residues out of 16 in the 65th repeat to just one residue within the 16 adjacent positions in the C-terminal domain.

## Table 1 | Critical number of INP repeats for ice nucleation activity

| Number of Repeats | T$_{50}$ (°C) - WISDOM | T$_{50}$ (°C) - BINARY |
|---|---|---|
| 0 (Buffer) | −38.7 | −29.7 |
| 12 | −38.3[†] | −28.2[†] |
| 15 | −23.4 | −22.9 |
| 16 | −21.3 | −22.5 |
| 17 | −19.8 | −19.4 |
| 18 | −17.5 | −17.4 |

The T$_{50}$ nucleation temperatures obtained using data from replicate WISDOM and BINARY trials are tabulated for *Pb*INP constructs containing 12 to 18 repeats. The temperature uncertainty for both the WISDOM and BINARY experiments is ±0.3 °C.
[†]These temperatures are not significantly different from the T$_{50}$ values of negative control experiments with LB and PBS on the WISDOM and BINARY, respectively.

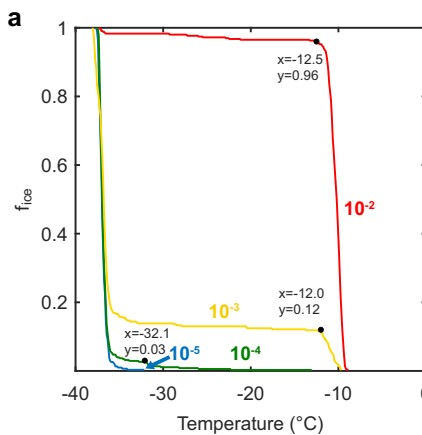
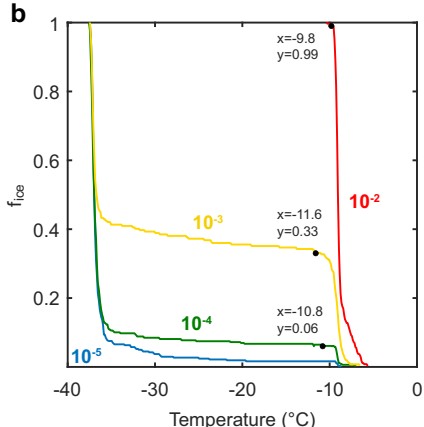

**Fig. 4 | Uniform ice nucleation by INP-expressing *E. coli* and *P. borealis*. a** *E. coli* expressing the 33-repeat *Pb*INP construct were assayed for the fraction of droplets frozen ($f_{ice}$) as a function of temperature for different dilutions of the starting culture on the WISDOM apparatus. The red, gold, green, and blue curves are for 100-fold, 1000-fold, 10,000-fold, and 100,000-fold dilutions, respectively. The curves are the mean of three measurements and the inflection points are marked with the transition temperature and the fraction of frozen droplets at that point. **b** Nucleation curves for cold-acclimated *P. borealis* that were diluted and treated identically.

The match to the consensus is particularly good in the first 54 of the 65 central domain repeats. Indeed, 45 of these 54 repeats have at least 12 residues matching the consensus. In contrast, none of the last 11 repeats have 12 residues matching the consensus. The consensus match weakens through the last 11 repeats adjacent to the C-terminal domain from a high of 11 to a low of 6 positions being conserved (Fig. 5a).

The degree of conservation within the more consistent 54 N-terminal repeats and the more variable 11 C-terminal repeats is depicted using WebLogo plots, illustrating sequence conservation through bit scores as well as the frequency of alternative residues at each position (Fig. 5b). Apart from locations 11, 12, and 14, most positions in the first 54 repeats are occupied by highly conserved residues, indicating regularity at the sequence level that likely results in conserved features of the central solenoid domain, including tandem arrays of the water-organizing TxT and SLT motifs, and the possible YGS dimerization motif[12]. In the WebLogo plot of the 11 C-terminal repeats, the putative water-organizing residues and the Y at position 3 are largely absent and a well-conserved R appears at position 12. The difference in solenoid coils between the first 54 that are water-organizing and the last 11 that generally lack water-organizing motifs is illustrated in Fig. 5c.

To gauge the functional importance of different regions of the central repeats of *Pb*INP, pairs of restriction sites were exploited to generate deletions of the central domain's N-terminal (*Sac*II), middle (*Eag*I), and C-terminal (*Pst*I) portions in each case deleting 32 repeats. Using the restriction nuclease *Eco*RI, it was possible to delete the 41-residue C-terminal domain. The four corresponding ice nucleation curves from BINARY for cultures bearing these deletions were compared to controls with full-length *Pb*INP (Fig. 5d). Ice nucleation activities of cultures with the N-terminal or middle portions of the central domain deleted follow the trend from Fig. 3, with a $T_{50}$ of −8.6 and −9.0 °C, respectively, compared to full-length cultures at a median of −7.1 °C. In contrast, deletion of the C-terminal portion of the repeat region, or the small C-terminal domain, essentially eliminated ice nucleation activity, lowering the $T_{50}$ to −27.1 and −26.4 °C, respectively, which is close to the phosphate-buffered saline $T_{50}$ of −29.7 °C in the BINARY apparatus.

### Mutagenesis of highly conserved INP motifs

Three constructs were produced to evaluate how ice nucleation activity was affected when distinct amino acid patterns in the central domain were mutated. The highly conserved **TxT**, SL**T**, and **Y**GS motifs are prominent in the β-solenoid model[12], which predicts that the threonine and tyrosine residues targeted for mutagenesis (in bold) is facing outward from the solenoid core (Figs. 1b, 6a). Outward-facing residues are the ones expected to participate in the water-organizing function of INPs, and mutation of these residues is unlikely to destabilize the core of the central *Pb*INP domain. To further minimize this risk of disrupting the protein fold, just a subset of motifs was mutated, and these mutations only occurred in repeats 18 to 40 of full-length *Pb*INP (Fig. 6b). Ice nucleation experiments produced freezing curves for the three motif mutants, which were compared to those of full-length *Pb*INP and the corresponding deletion variant (Fig. 6c). When a subset of T in the TxT or SLT motifs were mutated to S, K, or Y, culture ice nucleation activity was like that of the 42-repeat construct, where these central 23 coils were deleted instead of mutated. The WISDOM $T_{50}$ values of the TxT and SLT threonine mutants, at −10.5 and −10.1 °C respectively, were within the experimental uncertainty of one another, suggesting that both the TxT and SLT motifs contribute almost equally to the organization of water molecules. Cultures in which a subset of Y in the YGS motif were mutated to D or N resulted in a similar reduction in activity, as can be seen from its WISDOM $T_{50}$ of −10.5 °C. The overall conclusion from this set of experiments is that the background ice nucleation activity from the unmutated repeats in these constructs was sufficient to partially mitigate the impact of the mutations. To circumvent this problem, we moved the mutated regions into a shorter *Pb*INP construct where the mutations would not be masked by the high activity of unmodified repeats.

In these experiments, a total of 19 unmutated repeats flanked the mutations and confirmed the main conclusion that mutation was more deleterious than deletion, presumably because it interrupted the continuity of the wild-type water-organizing surface that remained. Also, the order of motif mutation severity was the same. Cultures of the construct with 19 uninterrupted repeats nucleated freezing at −16 °C on BINARY (Fig. 6d). When the SLT mutations in 23 repeats were inserted into the 11th repeat, ice nucleation activity fell by 2.5 °C to −18.5 °C. Insertion of the TxT mutations dropped the ice nucleation temperature by 5 °C to −21 °C and insertion of the YGS mutations pushed the ice nucleation temperature down by a total of 7 °C to −23 °C.

### Insertion of a bulky domain within the central domain of *Pb*INP

Given that mutation of the central repeats reduced the effectiveness of the remaining wild-type repeats by breaking their continuity, we explored the effect of a steric disruption on the continuousness of the tandem repeats and perhaps the ability of the INPs to dimerize[12]. This was tested here by the introduction of the fluorescent protein mRuby2, with a height of 4.2 nm and width of 2.4 nm (based on its near

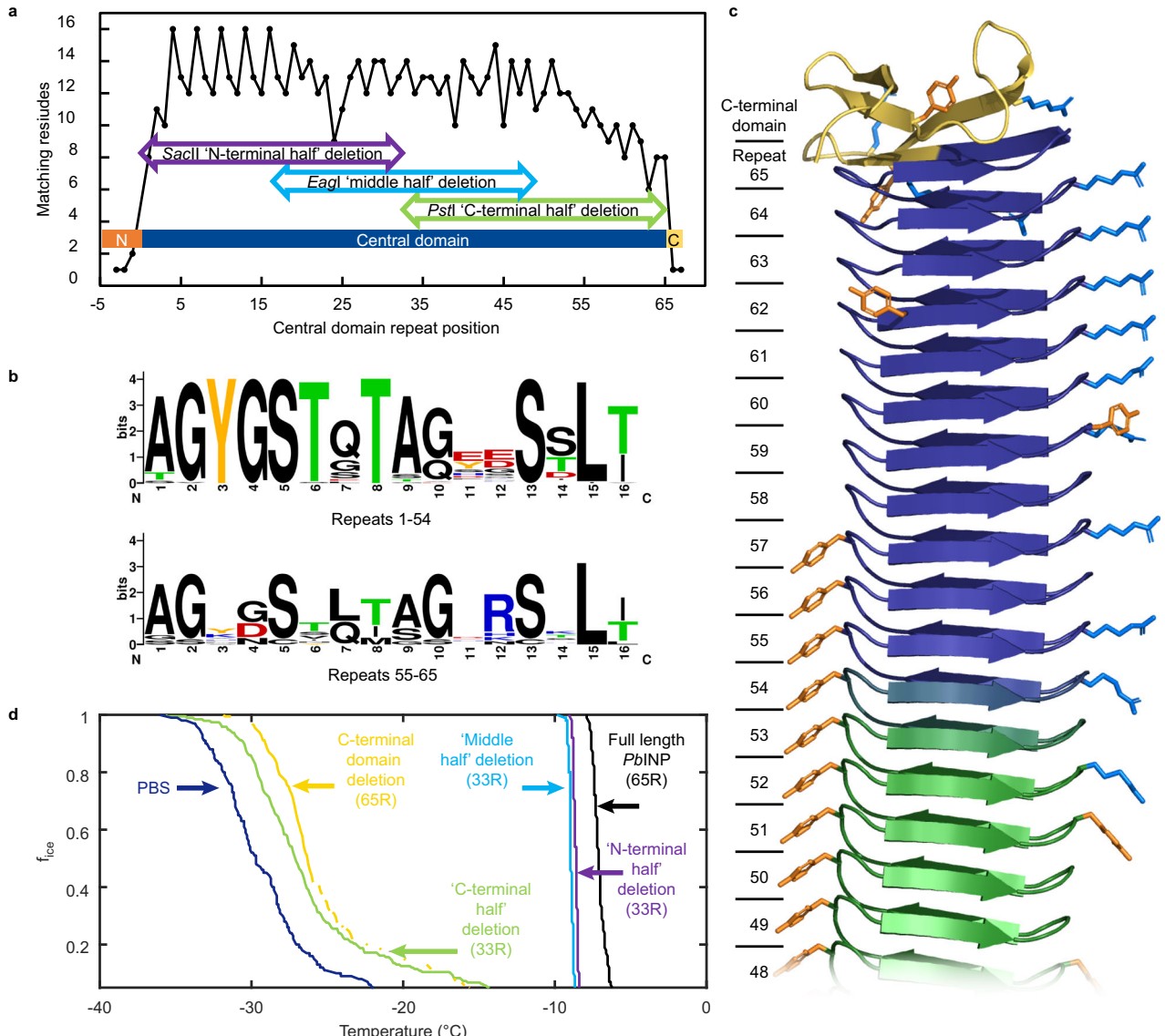

**Fig. 5 | Sequence conservation of the central tandem repeats in *Pb*INP.**
**a** Comparison of each numbered repeat in *Pb*INP, from the N to C terminus, to the consensus peptide sequence: AGYGSTQTAGEESSLT. Data points in black represent the number of matches to the 16-residue sequence, where a perfect match is 16. The narrow dark blue bar spanning the bottom of the graph represents the central domain. Repeats −3 to −1 compare the last 48 residues of the N-terminal domain to the 16-residue consensus sequence, while repeats 66 and 67 correspond to the first 32 residues of the 41-residue C-terminal domain. Horizontal arrows indicate the repeat region deleted in three different half constructs. **b** WebLOGO plots[73] of the first 54 and last 11 sixteen-residue repeats in *Pb*INP. Each stack's overall height indicates its degree of sequence conservation, while the height of a letter's symbol

refers to its proportional frequency at that position. **c** AlphaFold model of the C-terminal region of *Pb*INP. Beta strands are represented as arrows. The central repetitive domain is shown in dark blue, the C-terminal domain in yellow, the R side chains in light blue, and the Y side chains in orange. Repeats 49 to 65 are numbered from the N-terminal end of the central domain. **d** BINARY nucleation curves for three *Pb*INP constructs in which the front (purple), middle (blue), and back (green) halves of the central repetitive domain were deleted, leaving 33 repeats in each sample. The *Pst*I sites used to delete the back half were present in an earlier version of the synthetic gene and removed repeats 33–64 inclusive. The nucleation curves corresponding to the C-terminal domain deletion (yellow), full-length *Pb*INP (black), and the buffer control (dark blue) are also shown.

identity to GFP[39]), into the repetitive region of *Pb*INP, which has an estimated height of 2.2 nm and width of 1.0 nm. An advantage of using mRuby2 for this purpose is that its fluorescence indicates the functional production of the construct by the bacteria. Initially, the mRuby2 DNA was inserted into a 46-repeat construct rather than the 65-repeat construct to help accentuate any effects of the steric disruption that might have been muted in a full-length construct background. The insertion then replaced repeats 9 to 28, and a second construct re-positioned mRuby2 towards the C terminus by twelve INP repeats to replace repeats 21 to 40 (Fig. 7a).

Fluorescence microscopy images after growth and induction of *E. coli* transformed with mRuby2-*Pb*INP confirmed the expression of the

large mRuby2-*Pb*INP-GFP constructs, as evidenced by the yellow color of cells upon overlaying emissions from the red and green channels (Fig. 7b). Cultures of the first construct, designated R9-mRuby2-*Pb*INP due to insertion of mRuby2 within repeat 9, generated $T_{50}$ values of −9.2 and −8.6 °C on the WISDOM and BINARY, respectively (Fig. 7c). These values were close to those of the 33-repeat construct ($T_{50}$ of −9.2 and −9.0 °C). This observation suggested that ice nucleation activity might be correlated with the longest uninterrupted water-organizing surface in the central domain, which would be 37 repeats. This hypothesis was tested with the second mRuby2 construct (R21-mRuby2-*Pb*INP), where the bulky fluorescent protein was shifted towards the C terminus by twelve repeats, such that the largest

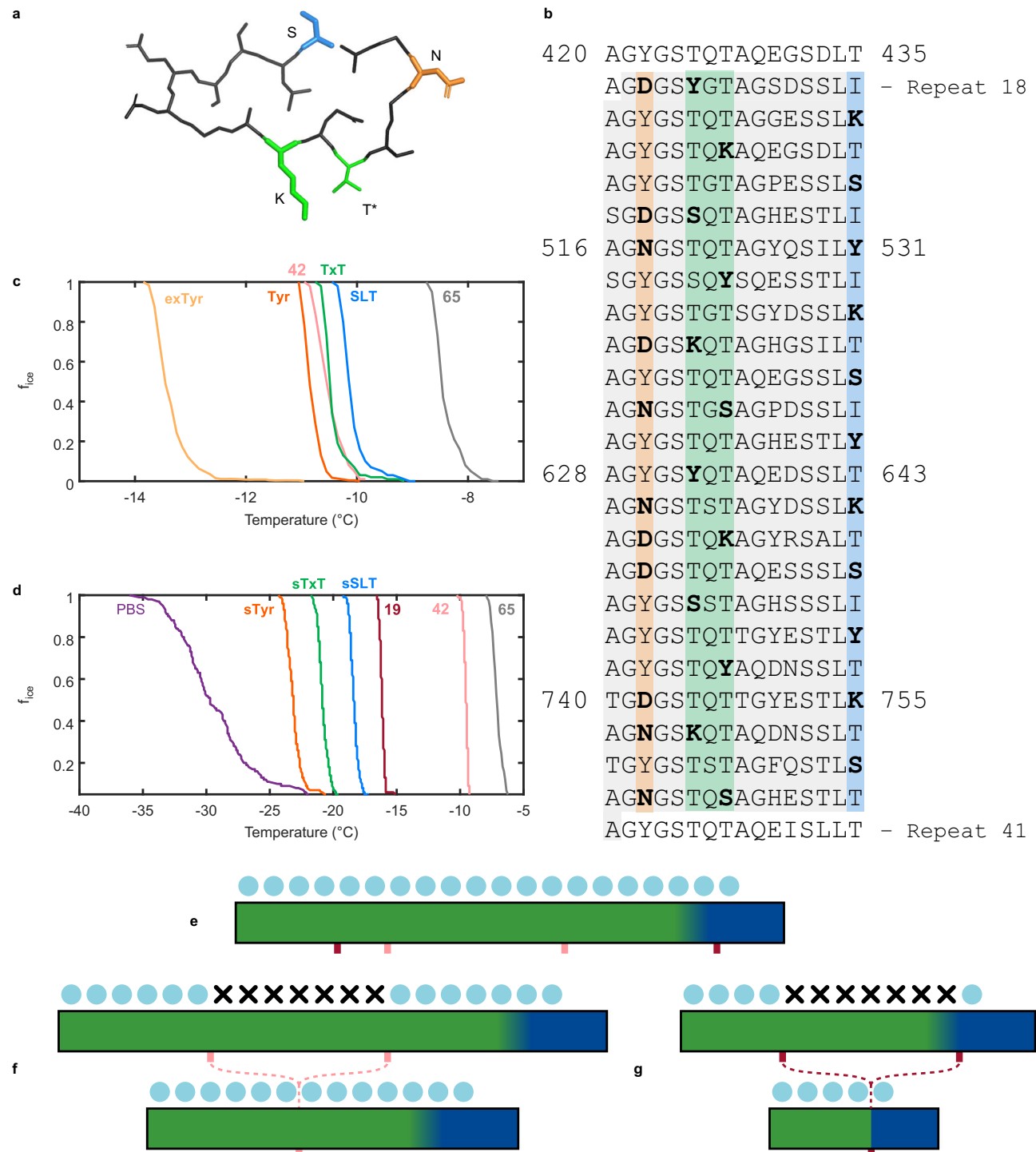

**Fig. 6 | Mutagenesis of highly conserved INP motifs. a** Solenoid cross-section showing examples of mutated residues in the three targeted motifs, where N is orange, K is green, and S is blue. T* indicates the other position where threonines were altered in the TxT mutant. **b** The *Pb*INP amino acid sequence from residues 420 to 819, based on numbering from the initiator methionine. Three constructs targeted distinct amino acid patterns: mutations are shown in bold in the tyrosine ladder (orange), threonines of the TxT motif (green), and threonines/isoleucines of the SLT motif (blue). The region shaded in gray represents the residues removed in the 42-repeat "*Pst*I deletion" construct. All mutated residues are located within the 23-repeat segment shown here. **c** WISDOM nucleation curves from full-length mutagenesis constructs with 65 repeats. From right to left, curves are colored gray (full-length *Pb*INP), blue (SLT mutant), green (TxT mutant), pink (mutated region deletion), and orange (Y ladder mutant). In gold is a slight extension of the Y ladder

mutations where an additional five Y flanking the 23 mutated repeats were changed to N or D (exTyr). **d** BINARY nucleation curves from constructs that have the same mutant fragments in a shorter central domain (sSLT, sTxT, and sTyr), such that 23 out of 42 repeats contain mutations. The color scheme is the same as in part C. The 19-repeat construct without any mutant fragment is shown in burgundy. On the extreme left is the negative control of PBS in purple. **e** Depiction of the full-length solenoid where the green and blue rectangles represent water-organizing coils and R-coils, respectively. Above the former are organized ice-like waters (blue spheres). **f** The full-length construct where the 23-repeat mutant fragments disrupt ordered water molecules, which are marked by Xs, and the equivalent constructs where the mutated region is deleted. **g** The second mutation series, where the 23-repeat mutant fragments were cloned into a shorter central domain for a total of 42 *Pb*INP repeats, and the equivalent shorter constructs where the mutated region is deleted.

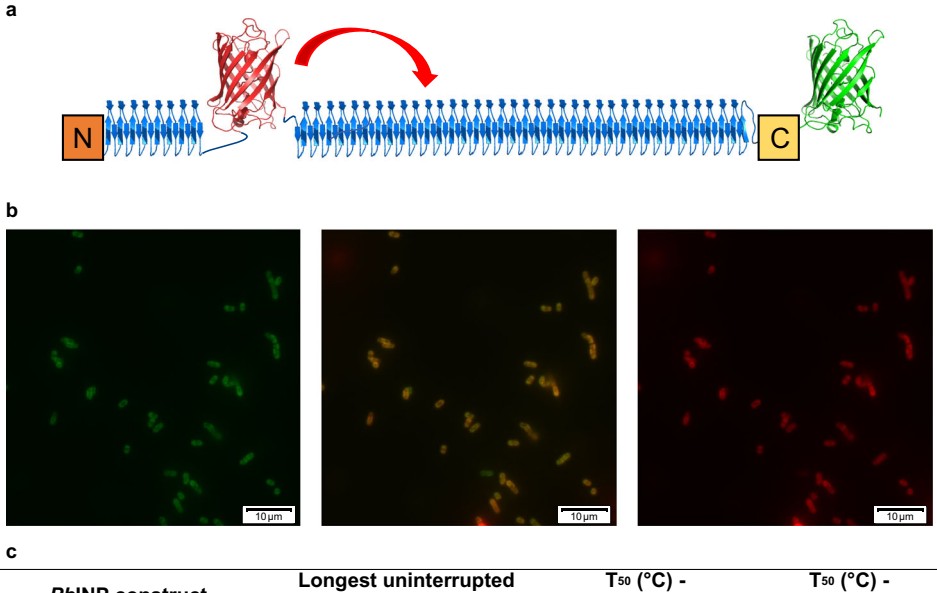

| PbINP construct | Longest uninterrupted number of repeats | $T_{50}$ (°C) - WISDOM | $T_{50}$ (°C) - BINARY |
|---|---|---|---|
| 65 repeats (full length) | 65 | -8.4 | -7.1 |
| 46 repeats | 46 | -8.9 | -8.5 |
| R9-mRuby2-PbINP | 37 | -9.2 | -8.6 |
| 33 repeats | 33 | -9.2 | -9.0 |
| R21-mRuby2-PbINP | 25 | -11.5 | -11.4 |

**Fig. 7 | Insertion of mRuby2 within the central domain of PbINP. a** An illustration of the R9-mRuby2-PbINP construct, where the large fluorescent protein replaces repeats 8 to 27. Domain colors are identical to those in Fig. 2. Known and modeled structures are presented in ribbon format. The red curved arrow depicts the re-positioning of mRuby2 towards the C terminus by twelve repeats in R21-mRuby2-PbINP. **b** Representative fluorescence microscopy images ($n = 3$, scale bar: 10 μm) of post-induction *E. coli* producing the R9-mRuby2-PbINP. The left and right panels were obtained after excitation at 488 and 560 nm, respectively. The middle panel is an overlay of these two images. **c** WISDOM and BINARY $T_{50}$ values for R9- and R21-mRuby2-PbINP, as well as the 33-repeat, 46-repeat, and full-length (65-repeat) constructs. The temperature uncertainty for both the WISDOM and BINARY experiments is ±0.3 °C.

continuous surface spanned 25 repeats. The WISDOM and BINARY $T_{50}$ values for R21-mRuby2-PbINP were −11.5 and −11.4 °C, respectively, which are 2.3 and 2.8 °C lower than those of the initial R9 construct. This shift to lower ice nucleation temperatures shows that the continuity of the water-organizing surface is important for potent ice nucleation activity.

## Discussion

AlphaFold predicted structures of the N- and C-terminal PbINP domains for which we were unable to find structural homologs in the database. The sequence for the globular fold at the front half of the N-terminal domain is conserved in INPs from other species and strains of ice-nucleating bacteria. The putative unstructured region that follows is also a common feature of these proteins, although differing in length and sequence. It is tempting to speculate that this long linker region might offer flexibility during the assembly of an INP superstructure or aggregate that is thought to be needed for ice nucleation at sub-zero temperatures as high as −2 °C[1,6,28,40].

AlphaFold structure prediction for the C-terminal domain of PbINP suggests that it forms a cap over the C-terminal end of the central solenoid repeats that covers the hydrophobic core of the solenoid and engages the beta strands of the last coil. Capping structures are essential for the stability of beta-solenoids[15,41], and without them, the solenoid tends to unravel or form amyloid fibrils[42]. The complete loss of ice nucleation activity seen with the deletion of the PbINP C-terminal domain is consistent with previous findings in the literature on inaZ[35] and *Erwinia ananas* INP[43]. Indeed, AlphaFold predicted an almost identical structure for the C-terminal domain of InaZ to that of PbINP (Supplementary Fig. 3). What we have added here is the definition and boundaries of the capping structure and the

discovery of the adjacent R-coils that are equally important for ice nucleation activity.

An advantage of having a synthetic INP gene with many restriction site pairs built in as silent mutations is that a wide range of deletions of varying length and location can be systemically generated. As a result, our deletion analyses have expanded on those conducted by ref. 27, who removed up to 34 INP repeats from *inaZ*, causing a 6 °C reduction in ice nucleation temperature from −3.7 to −9.7 °C. Larger central domain deletions were not produced in their study. Indeed, the lower detection limit of −13 °C on their apparatus would have restricted their measurements. In contrast, we measured more extensive PbINP deletions on the WISDOM and BINARY devices, which can evaluate freezing events at much lower temperatures approaching the homogeneous freezing point.

In cultures of deletion constructs with only 18 repeats remaining, the ice nucleation temperature was −17 °C. The deletion of a further six repeats caused a complete loss of activity to the homogeneous ice nucleation point (about −38 °C in the WISDOM apparatus). We were then able to interrogate this differential by insertion of oligonucleotides that encoded three, four or five repeats. The restoration of three repeats to a total of 15 boosted the ice nucleation temperature to −23 °C. Between 15 and 18 repeats, a single repeat addition was associated with a 2−3 °C increase in $T_{50}$, indicating a steady restoration of activity near the "critical number" of repeats in larger increments than temperatures measured at the upper end of the range. Indeed, a 2−3 °C change in $T_{50}$ was barely attained on the WISDOM and BINARY instruments when the number of central domain repeats was halved from 65 to 33.

Structure predictions done on PbINP using AlphaFold in combination with the WEBLOGO plots indicate that the C-terminal 11 coils of

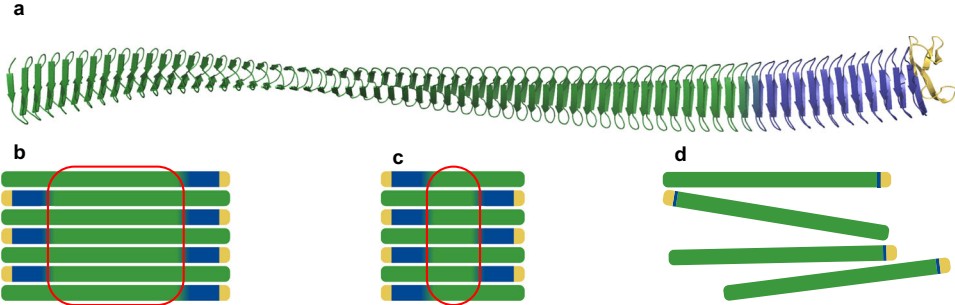

**Fig. 8 | Model for the involvement of R-coils in INP multimerization. a** AlphaFold model for *Pb*INP without its N-terminal domain. Water-organizing coils are shown in green, R-coils in blue, and the C-terminal domain in gold. **b** With full-length INPs, interactions between R-coils and water-organizing coils enable the formation of a large, wide surface. **c** A decrease in water-organizing coil numbers reduces the length of the ice-nucleating surface. **d** Loss of a critical number of R-coils leads to a breakdown in multimerization.

the central repeat region appear to diverge in sequence and presumably also diverge from an ice nucleation function. Notably, the previously well-conserved water-organizing motifs are largely absent from this portion of the solenoid. Instead, the appearance of a regular array of R at position 12 in the coil suggests another function for these repeats (R-coils), which we speculate could be associated with the formation of a patch of INPs. This would appear to be critical because deletion of the C-terminal 33 repeats from the central core eliminated all ice nucleation activity, whereas the loss of the same number of repeats from the center or the N-terminal end of the central domain resulted in only a minor reduction in ice nucleation temperature. In one model (Fig. 8), the R-coils (in dark blue) make side-by-side interactions with the green water-organizing coils as part of INP oligomerization. These connections could be electrostatic between the stacked arginines of the R-coils and the glutamates and aspartates that occupy the equivalent position in the water-organizing repeats. There are several reports that physiological pH and electrostatic interactions are important for the assembly of INP multimers[44-46]. Modest shortening of the water-organizing coils would not stop the ability of the INPs to associate, hence the relatively minor reduction in ice nucleation temperatures for aggregates of *Pb*INPs with partially deleted water-organizing regions explored in our study. Shortening the water-organizing coils below the length of R-coils could weaken the assembly, whereas deletion of the R-coils would eliminate the possibility of patch formation.

Thus, in addition to the total number of repeats, it is important to consider how many of these can potentially organize water. The *Sac*I construct with its 18 solenoid coils has only ten of those "water-organizing repeats" while the other eight are C-terminal R-coils. Therefore, the high ice nucleation temperature of such a short functional construct is surprising and must surely entail INP multimerization. Similarly, the complete inactivity of the 12-repeat *Spe*I construct is consistent with this interpretation since it only has 3 water-organizing repeats, fewer even than any solenoidal AFP, and too few to form a stable interaction with the R-coils. It is thus remarkable that the addition of three, four, and five more water-organizing repeats was able to restore some ice nucleation activity (Table 1), which again argues for the formation of a network of these proteins rather than them acting independently. However, the network of INPs may not adopt the arrangement depicted in Fig. 8: similarly plausible models could involve INPs in a parallel orientation or a three-dimensional bundle rather than a perfectly flat sheet. The uncertainties with these models include the effect of the twist in the solenoid and whether the R-coils can interact with any segment of the water-organizing coils. These possibilities could lead to branching and extension of the INP network through a staggered assembly. It should be noted that the AlphaFold model predicts less than 3 degrees of rotation between adjacent coils; this relatively small amount of twist may be eliminated by the

zippering together of neighboring solenoids to form a flat surface without unduly straining the structure. Recently, an InaZ deletion construct without its N-terminal domain was expressed in *E. coli*, leading to cytoplasmic INP retention, yet the ice nucleation temperature was only 2–3 °C lower than that of full-length InaZ[47]. This suggests that large INP assemblies were able to form in the cytoplasm and efficiently nucleate ice without a membrane for support. Thus, unanchored INP aggregates in solution might not produce the flat surface depicted in Fig. 8.

Recently, a soluble INP construct with a central domain length in the lower range of our experimental work was reported. This is a 16-repeat *P. syringae* construct purified by size-exclusion chromatography, which had $T_{50}$ values that gradually diminished from −10 to −20 °C with decreasing INP concentrations, as measured using a freezing array similar to the BINARY apparatus but without individual droplet compartmentalization[48]. The expression of an active and purifiable repeat-truncated INP is an accomplishment, although it is important to note that INP activity and higher-order assembly may be different in vitro in the absence of a bacterial membrane. An even shorter construct of six central INP repeats was reported to show ice-binding activity in vitro but was not tested for ice nucleation[49].

The consistency in our in vivo expression protocols, combined with "blind" measurements employing two independently established ice nucleation apparatuses, add rigor to our results and enable reliable comparisons between *Pb*INP variant activities. This is particularly relevant for the plot of $T_{50}$ values against central repeat length (Fig. 3c), which shows minor changes in $T_{50}$ values up to the point where 30+ repeats were deleted from the central domain, followed by a steep but continuous reduction as $n_{rep}$ decreased from 29 to 12. We can now genetically fine-tune a molecular ice nucleator to a level that allows ice nucleation at any desired temperature between the highest $T_{50}$ of the 65-repeat wildtype and the complete loss of ice nucleation for the protein shortened to 12 repeats. Moreover, we show that the same effect can be achieved by introducing an interruption to the continuous water-organizing surface at any desired place in the 65-repeat protein, confirming that the largest number of continuous repeats is responsible for ice nucleation. With this knowledge, the ice nucleation temperature of water samples (both small and large) can be predetermined in a rational manner with high accuracy.

In designing the mutagenesis constructs, we were guided by similarities between recurrent motifs from some structurally characterized AFPs and conserved residues in the central domain of *Pb*INP. For example, the TxT motif appears in the same location in all coils of the INP solenoid model and would form a tandem parallel array indistinguishable from the ice-binding sites of insect AFPs like those from *Tenebrio molitor* and *Dendroides canadensis* beetles[17,18] and spruce budworm[16] except in its much greater length. To evaluate the role of the TxT motifs in ice nucleation, we used the same mutagenesis

strategy that established their ice-binding role in AFPs. Threonine residues at intervals along the tandem parallel array were replaced by Y, S, and K to disrupt the flatness and water-organizing ability of this surface while being compatible with the underlying beta-sheet structure of the solenoid. T to Y mutations were previously effective in delineating the ice-binding site of *T. molitor* AFP[20] and the bacterium, *Marinomonas primoryensis*, ice-binding protein[50]. T to S mutations, which remove the crucial hydrophobic methyl group, were effective in knocking out the ice-binding activity of type I AFP[51–53]. Mutations to K also successfully knocked down the activity of another type I AFP[54].

The SLT motif is present on the opposite side of the modeled β-solenoid surface, which is also flat. In the model of the INP dimer[12], where the second INP lies parallel but is 180° rotated, this motif can be viewed as a lateral extension of the TxT motif to produce a wider water-organizing surface. This resembles the ice-binding sites seen in some beetle and moth AFPs that have four[55,56] or five[57] parallel ranks of T. The first and third positions of this SLT motif are less well-conserved than those in the TxT sequence, which is also observed in the ice-binding site of the grass *Lolium perenne* AFP[58]. That the $T_{50}$ values of the TxT and SLT mutants in full-length constructs are comparable in activity to the deletion of the corresponding region suggests that their ability to serve as templates for ordered arrays of water molecules has been completely disrupted (Fig. 6e–g). However, direct comparisons are difficult since deletion of the 23 repeats leaves 42 continuous repeats intact, whereas mutations in the 23-repeat segment potentially split the water-organizing region into two much smaller zones of 18 and 24 repeats on the N- and C-terminal sides, albeit the 24 repeats on the C-terminal side include 8 or more coils that are likely not water-organizing repeats. To better evaluate the effects of mutations interrupting the continuity of the water-organizing region, we moved the mutated region into a shorter construct where the sensitivity of the WISDOM and BINARY instruments was better (Fig. 6d). This approach clearly showed that the loss of ice nucleation activity from mutating the water-organizing motifs was made worse by interrupting the continuity of the remaining water-organizing region. This result was confirmed by the mRuby2 insertion experiments that also broke the continuity of the water-organizing repeats. Furthermore, although GFP has been fused to the C terminus of InaK[59], InaQ[60], and to *Pb*INP or *Pb*INP with a deletion of the internal repeats[61] to perform imaging or subcellular localization experiments, our two mRuby2-*Pb*INP constructs demonstrate functional protein insertion within the central domain of an INP. Fluorescence microscopy images confirmed successful expression of the C-terminal GFP and the internal mRuby2 tags (Fig. 7b), suggesting that the *Pb*INP function is surprisingly resilient to both repeat deletions and a bulky insertion in the central domain.

In both the mutagenesis and mRuby2 insertion, we observe a clear pattern: the number of uninterrupted repeats determines the $T_{50}$ value. The original rationale for perturbing the YGS sequence by mutagenesis was to interfere with a putative dimerization motif. According to the Garnham model[12], the antiparallel *Pb*INP dimerization through the tyrosine ladder places the TxT β-strand of one monomer on the same face as the SLT from the other monomer to increase the water-organizing surface area. In AFPs, even a small increase in the water-organizing surface leads to greatly increased ice-binding efficiency as measured by freezing point depression[62–64]. Thus, we anticipated a comparable enhancement in ice nucleation activity. The outward-pointing tyrosine residues help form tight turns between the two beta-sheets and were replaced here at irregular intervals with D or N, two residues that can serve this function while interrupting the stacking of the aromatic side chain that is required for dimerization. However, an AFP from midges has been found with a beta-solenoid fold that has an ice-binding site made up of a row of seven stacked Y. Mutation of the central Y to a D was sufficient to eliminate antifreeze activity[65,66]. Therefore, it is possible that the reciprocally stacked Y in *Pb*INP could serve either a dimerization function or as a water-

organizing surface, where only some Y are engaged in dimerization. Furthermore, the AlphaFold structure predicts that the S of the YGS sequence points inwards to form a serine ladder at position 5 equivalent to the serine at position 13 (Fig. 1b, c), making the solenoid coil more symmetrical with the beta strands parallel to each other, rather than offset at an angle. We speculate that the position of S, either inward or outward, could result in interchangeable isomers functioning in both water organization and dimerization.

Insertions, deletions, and mutations of *Pb*INP could potentially destabilize the protein fold and contribute to a decrease in activity. When we used AlphaFold to predict the structure of shortened or mutated constructs, there was essentially no change in the protein structure. It would be ideal if the fold of modified INPs could be assessed using IR and/or CD spectroscopy, but it has not yet been possible to isolate pure native INPs or their assemblies. Instead, their ice nucleation activity has been measured within the host bacterium in the presence of thousands of other proteins that would mask the IR or CD signals from the INPs. However, we note that none of the mutation sets caused a catastrophic loss of activity or a failure to fold the C-terminal GFP domain, which acts as an internal control for folding. This result is consistent with the mutated T and Y in the three water-organizing motifs being on the outside of the solenoid rather than contributing to the protein core, which in turn supports previous INP structural models[12,13]. Our experiments have advanced the understanding of INP function. The loss of INP activity from mutations of the TxT and SLT motifs are consistent with a putative water-organizing role. The slightly larger loss of ice nucleation activity from mutating the YGS sequence could be consistent with a dimerization function and water organization. Regardless, our results underscore the importance of the TxT, SLT, and YGS sequences functioning in concert for efficient INP-induced ice formation at high sub-zero temperatures. Taken together, these mutation experiments strongly support the idea that INPs organize water molecules much the same way as AFPs but on a larger scale, afforded by their much longer length and ability to associate into aggregates, resulting in a distinct function of nucleating ice.

## Methods

### AlphaFold model of *Pb*INP

The structure of *Pb*INP was modeled using AlphaFold, an artificial intelligence program designed to predict protein structure[67]. A slightly simplified version of AlphaFold v2.0 was accessed using a Colab notebook (Colabfold) maintained by AlphaFold's designers; the translated *Pb*INP gene was used as the input sequence. To compare the β-solenoid coils, a 16-residue section matching the consensus sequence for repeats 1–54 (AGYGSTQTAGEESSLT) was isolated from both the AlphaFold model and the originally proposed *Pb*INP β-solenoid model[12]. This comparison and the predicted C-terminal region fold were visualized using PyMOL v2.5.0 (Schrödinger, LLC).

### Production of *Pb*INP constructs

**Synthesis of *Pb*INP genes.** The gene encoding the *P. borealis* INP (GenBank accession number: EU573998) was altered to introduce silent mutations that generated useful restriction sites. To produce internally truncated *PbINP* constructs, we confirmed that pairs of introduced restriction sites were unique, with the sites appearing only twice in the synthetic *PbINP* and not elsewhere in the pET-24a vector backbone. Furthermore, restriction sites were introduced at equivalent points in the repeats such that they maintained the periodicity of the central repetitive domain. For example, an internal truncation would join position four of a 16-residue repeat to position five of some later repeat. The DNA sequence for enhanced green fluorescent protein (GFP; GenBank accession number: AAB02572) was ligated to the 3′-end of the *PbINP* sequence via a hexanucleotide encoding two linker residues (Asn-Ser). The entire DNA sequence was codon-optimized for expression in *Escherichia coli* without disrupting any planned silent

mutations or restriction sites and synthesized by GenScript (Piscataway, NJ). This construct was sub-cloned into pET-24a and sequenced in both directions. As with all constructs produced by GenScript, a certificate of analysis confirmed that the correct DNA sequence had been synthesized and successfully cloned into the expression plasmid. Biological materials produced and used in this study can be obtained from the corresponding author.

### Formation of deletion construct vectors
The plasmid was transformed into chemically competent *E. coli* Top10 cells (Invitrogen) according to the manufacturer's instructions. The production of *Pb*INP deletion constructs followed identical steps, differing only in the restriction endonuclease used for plasmid digestion. Plasmid DNA, encoding full-length *Pb*INP, was purified from overnight culture using the GeneJET Plasmid Miniprep Kit (Thermo Scientific). Eluted full-length *Pb*INP plasmid DNA was mixed with the applicable restriction enzyme (New England Biolabs) and buffer for digestion, then subjected to agarose gel electrophoresis. The larger *Pb*INP vector fragment (~8500–6500 bp) was excised from the DNA agarose gel and purified using the QIAquick Gel Extraction Kit (Qiagen) before overnight incubation with T4 DNA ligase. *E. coli* Top10 cells were transformed with 1–2 μL of ligation product to produce individual colonies that were grown overnight in Luria broth (LB) and kanamycin. Deletion constructs were then purified using the miniprep kit and analyzed by restriction digest and gel electrophoresis. Sanger sequencing (Robarts Research Institute, London, Ontario) confirmed deletions.

To generate constructs with 15, 16, and 17 repeats, a 288-bp synthesized DNA (Supplementary Table 1) encoding six 16-residue repeats, with pairs of unique restriction sites 48, 96, and 144 bp apart, was inserted into the 12-repeat *Pb*INP deletion construct (GenScript). Internal deletions to generate 15-, 16-, and 17-repeat constructs were done as described above.

### Mutation of putative water-organizing motifs and dimerization surfaces
In producing the central domain mutations in the TxT and SLT motifs and tyrosine ladder, it was important to strike a balance between adequately mutating conserved residues and avoiding large-scale disruptions to the overall fold of the repetitive region. To reduce the possibility of misfolding, only a subset of repeats was mutated and at fewer than half of the motif positions. Also, the introduction of charged residues on adjacent repeat coils was generally avoided. Furthermore, to facilitate comparisons between the constructs and, therefore, the contributions of the targeted motifs to ice nucleation activity, mutations in all three constructs were restricted to repeats 17 to 39, which corresponds to the segment removed in a deletion construct that left 42 repeats. The region containing mutated DNA was synthesized for each of the targeted motifs, then individually sub-cloned into full-length *Pb*INP by GenScript to produce the three mutagenesis vectors.

To enhance the effects of these motif disruptions, a separate series of mutant constructs was produced by moving the mutated 23-repeat segments from the initial full-length *Pb*INP vectors and inserting them into a 19-repeat *Pb*INP construct. The latter was made by ligating annealed complementary 48-base oligodeoxynucleotides with *Sac*I overhangs (Supplementary Table 1) into the 18-repeat *Sac*I deletion that then formed the 19th repeat. The 48-base segment contained a unique restriction site to receive the 23-repeat mutated region without changing the periodicity of the repeats. Mutant constructs in this latter series each have a total of 42 repeats, of which 23 contain the mutations.

### Insertion of mRuby2 into *Pb*INP
The R9-mRuby2-*Pb*INP construct, which has mRuby2 inserted within the ninth central tandem repeat, was produced through PCR amplification of *mRuby2*, followed by restriction digestion and ligation into the 46-repeat *Pb*INP deletion construct. Thus, the fluorescent protein replaces the segment between bases 899 and 1811 (repeats 9 to 28), thereby splitting the 46 repeats in the central domain into two uninterrupted sections with 9 and 37 repeats. Similarly, the R21-mRuby2-*Pb*INP construct was of the same length but had mRuby2 inserted in the place of bases 1475 and 2387 (repeats 21–40), which splits the central domain into 21 and 25 repeats. This latter construct was synthesized by GenScript.

### Ice nucleation experiments
**Transformation of *E. coli* Arctic Express.** The activity of each *Pb*INP construct was measured in the Arctic Express (DE3) strain of *E. coli* (Agilent, Santa Clara, California) since its constitutive expression of two cold-adapted chaperones promotes correct folding of proteins at low temperatures[68]. Electrocompetent cells were mixed and incubated on ice with constructed *Pb*INP vector, then the solution was pipetted into a chilled cuvette and subjected to electroporation according to the manufacturer (Bio-Rad). After the addition of 500 μL LB, cells were allowed to recover at 37 °C for 1 h and were then spread onto LB-agar plates.

### Culturing of bacteria
All *E. coli* cultures were grown in LB supplemented with 25 μg/mL of gentamicin and 100 μg/mL of kanamycin at 37 °C with shaking at 235 rpm until the culture reached an $OD_{600}$ of ~0.6. Expression of *Pb*INP variants and the full-length construct was induced using 1 mM isopropyl β-d-1-thiogalactopyranoside (IPTG) overnight at 10 °C with shaking at 235 rpm. Bacterial samples were stored at 4 °C for 2 days prior to ice nucleation measurements to potentially encourage the formation of higher-order INP structures[69]. All samples were analyzed for ice-nucleating activity in a "blind" manner: bacteria were shipped in tubes marked only with a sample number, and the genetic identity of each *Pb*INP construct was revealed after ice nucleation measurements had been performed.

Those samples not assayed on the day of preparation were cryopreserved in 15% glycerol after reaching an $OD_{600}$ of ~0.6 and stored in 1.5-mL vials at −80 °C. When these samples were subsequently assayed, the cultures were thawed at 37 °C in a water bath for 4 min, followed by three wash steps with LB broth and centrifugation to remove the glycerol before being restored to their original volume in LB. There were no statistically significant differences in the ice nucleation temperatures of fresh versus previously frozen cultures.

### Bacterial dilution to single nucleators
Serial dilutions of cultures expressing *Pb*INP were achieved using identical protocols. After centrifugation and determination of the mass of the resulting cell pellet, bacteria were resuspended in sterile phosphate-buffered saline (PBS; 137 mM NaCl, 2.7 mM KCl, 4.3 mM $Na_2HPO_4$, 1.4 mM $KH_2PO_4$) to 1 mg/mL. An aliquot (100 μL) of this suspension was diluted in 9.9 mL of PBS and thoroughly mixed, yielding a $10^{-2}$ cell dilution, with subsequent dilutions of $10^{-4}$, $10^{-6}$, and $10^{-8}$ prepared. A 100-μL sample from each of these dilutions was spread onto separate solid media plates, which were incubated at 10 °C. The number of colony-forming units (CFU) was determined after overnight growth.

### Ice nucleation measurements in the WISDOM apparatus
Freezing experiments on suspensions containing live bacteria were performed using the Weizmann Supercooled Droplets Observation on a Microarray (WISDOM) apparatus[70,71]. Briefly, ~120 nanoliter-sized droplets (80–100 μm diameter) containing bacteria at a concentration of 1 mg/mL were generated in a microfluidic device in a narrow junction pressed by an oil phase and subsequently trapped in an array of chambers. The microfluidic device was then placed on a commercial

temperature-controlled stage (Linkam, LTS420), with the freezing of the individual droplets tracked by their brightness using an optical microscope (Olympus, BX51) equipped with a charged-coupled device (CCD) camera. Homogeneous ice nucleation of pure water droplets was observed below −35 °C, at a cooling rate of 1 °C per min, in agreement with theoretical predictions. The temperature uncertainty of the apparatus was calculated at ±0.3 °C. The frozen fraction $f_{ice}$, representing the accumulated fraction of frozen droplets at each temperature, was derived for each sample. $T_{50}$, the temperature at which 50% of the droplets are frozen (i.e., $f_{ice} = 0.5$), was used to compare different samples. Ice nucleation experiments were repeated twice using different microfluidic chips. The period between each repetition was up to 40 days, and the samples were cryopreserved during the time gap between the measurements as described above in subsection "Culturing of bacteria".

## Ice nucleation assays in the BINARY apparatus

The Bielefeld Ice Nucleation ARraY (BINARY)[72] apparatus was used for ice nucleation experiments with freeze-dried *Pb*INP-expressing bacterial cultures that were reconstituted with double-distilled H$_2$O. For each sample, three separate replicates were performed using ~64 droplets per trial. Droplets (0.6 μL) were placed on a hydrophobized glass plate and separated by a polydimethylsiloxane spacer, then they were cooled at a rate of 1 °C per min using a commercial cooling stage (Linkam, LTS 120). The droplets' freezing temperatures were detected by a change in the average gray value in the image of each droplet, observed with a CCD-Camera (Q-Imaging, MicroPublisher). Any droplets with direct contact with the device's polydimethylsiloxane spacer were not considered because of the potential for heterogeneous ice nucleation of this surface, and hence an effect on the freezing temperature of the droplets cannot be excluded. The apparatus was calibrated in terms of the absolute temperature and the rate of cooling. The overall uncertainty in the reported freezing temperature is ±0.3 °C.

The freeze-dried *E. coli* cultures were filled to their original volume by adding double-distilled water, and the resuspended samples were tested directly on the BINARY apparatus as well as samples diluted 100-fold in phosphate-buffered saline (PBS). As references for the undiluted and diluted experiments, respectively, we prepared an aqueous sucrose solution with a concentration of 10% weight/volume and diluted a portion of this solution with PBS buffer by a factor of 100.

For the evaluation of the freezing events, we used the same definitions of $f_{ice}$ and $T_{50}$ as those described for the WISDOM experiments above. The $T_{50}$ value obtained for pure double-distilled water on BINARY is −29.5 °C. This temperature can be regarded as the lower limit for heterogeneous freezing experiments of samples in pure water measurable in this apparatus (without the colligative effects of solutes).

## Comparison of *Pb*INP tandem repeats to a consensus sequence

The number of consensus-matching residues in each repeat was determined to assess sequence conservation across the central domain. A plot of the number of consensus-matching residues (out of a maximum of 16) for each repeat was produced using the sequence comparison output from Molecular Evolutionary Genetics Analysis (MEGA) software. The amino acid sequence for the last 48 residues of the N-terminal domain, the 65 sixteen-residue central domain repeats, and the first 32 residues of the C-terminal domain were compared to the consensus sequence AGYGSTQTAGEESSLT. The "Distance" analytical tool was used to "compute pairwise distances" using the "No. of differences" model to output the number of divergences per 16 residues. This number, subtracted from 16, is equivalent to the number of consensus-matching residues. Using a data analysis program, these values were plotted against the repeat position, where the N-terminal domain residues correspond to repeats −3 to −1, the central domain is represented by repeat positions 1–65, and the first 32 C-terminal domain residues are designated "repeats" 66 and 67.

## Fluorescence microscopy

Fluorescence microscopy images of fluorescently tagged *Pb*INP constructs were obtained using an inverted microscope (Olympus IX83, Germany) equipped with a high-speed CMOS camera (Zyla-4.2-CL10, ANDOR, Ireland). Post-induction *E. coli* cultures were imaged at 100x magnification in both the green and red fluorescence channels to detect GFP and mRuby2 using emission ranges of 495–540 nm and 570–625 nm, respectively. Overlaid images were produced using the IX83 microscope's associated software.

## Reporting summary

Further information on research design is available in the Nature Research Reporting Summary linked to this article.

## Data availability

The ice nucleation assay and AlphaFold model data generated in this study, alongside the synthesized *Pb*INP gene sequence, have been deposited in the figshare database at https://doi.org/10.6084/m9.figshare.20341440.v1.

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

## Acknowledgements

This work was in part funded by the CIHR Foundation award FRN 148422 to P.L.D., who holds the Canada Research Chair in Protein Engineering. A.B. acknowledges funding by the Planning and Budgeting Committee of the CHE in Israel, I.B. is grateful to the Israel Science Foundation (ISF 930/16), and V.K.W. acknowledges funding from NSERC Discovery Grant RGPIN-13289. We are thankful for the scientific contributions and technical expertise of Heather Tomalty and Rob Eves.

## Author contributions

P.L.D., I.B., C.G.B., and J.F. conceived and designed the experiments based on the *P. borealis* INP characterized by V.K.W., C.G.B., and J.F. prepared *E. coli* bearing the various INP constructs. J.F. prepared cultures for BINARY analyses that L.E. directly assayed. J.F. prepared bacteria for WISDOM analyses, which were cultured by A.B.; N.R. and A.B. measured these samples. T.H. performed the INP modeling. N.R., J.F., T.H., and L.E. produced the Figures. T.K., I.B., Y.R., and P.L.D. directed the study. P.L.D. and J.F. wrote the manuscript with input from V.K.W., T.K., I.B., and Y.R. All authors approved the final version of the manuscript.

## Competing interests

The authors declare no competing interests.
