## [Peer Review File · Nature Communications]

REVIEWER COMMENTS

Reviewer #1 (Remarks to the Author):

This manuscript by Forbes et al. reports an extensive set of experiments with synthetic ice nucleation proteins (INPs) to unravel the role of the size of the ice nucleation sites as well as the function of the N- and C-terminal structure. INPs of *Pseudomonas borealis* and mutants thereof are prepared and analysed with a whole array of ice nucleation assays. The secondary structure of the INP is predicted using the AlphaFold algorithm.

The article touches upon one of the most important and fascinating questions about INPs: What is the key driving force for ice activity? What is the role of the terminal domains? What is driving the formation of effective ice nucleating sites by assembly of INPs at the cell surface?

The authors draw a new picture of the molecular mechanisms: They identify new folding motifs in silico and relate them convincingly to experimental data. Protein engineering - truncation, blocking of site - is used as an effective experimental tool to test the hypothesis of the authors. The models are mostly supported by the data (see comments below).

This is a timely study and will be interesting to read for the large ice nucleation community but also for the broad readership of Nature Communications with a general interest science. The artwork is of high quality and the manuscript is structured well.

I recommend the article for publication after a the comments below are addressed.

1. The authors predict a solenoid structure for the INP. IN the antiparallel orientation within the assemblies this means that the relative orientation of the R-coil changes with respect to the neighbouring coils. This should affect the ability to aggregate and maybe even the shape of the formed complexes. The authors should comment on this.

2. The Garnham model proposed lateral assembly into large INP patches and the formation of a joint IN surface. This is also suggested in Figure 8. How is the formation of a flat area working when the assembly consists of twisting solenoids?

3. Self assembly of the INPs in Nature takes place at a cell surface. How does this enter the model? A flat model is drawn in Figure 8 but is this a likely scenario in solution?

4. Deletion of the C-terminal domain reduces ice activity. Is this explained by denaturing of the INPs?

5. Deletion of domains, particularly at the termini, or addition of bulky groups can severely destabilise the folding of the modified INP relative to what was predicted by AlphaFold. This can be an additional factor for the ice activity. Ideally one would measure the secondary structure of the constructs at least with IR or CD to make sure there are no significant changes. A structure prediction with AlphaFold could also be a good way, even though I have to admit I am not sure if the effort would be reasonable. At least the possibility should be mentioned in the discussion.

Recommendation: This paper is not recommended for publication at Nature Communications

The manuscript submitted by Forbes *et al.* examines the working mechanism of INPs by studying the effect of modifications of the size of the ice-nucleation site of the INPs. The reported experiments were done carefully and the manuscript is well written. The topic of the manuscript is important and it contains/confirms several interesting findings. The obtained results have however largely been known/published and are not sufficiently novel to be of interest to the broad *Nature Communication* audience, and may be better suited for a more specialized journal. In addition, there a few key points that need to be addressed prior to publication in any journal.

I first summarize the key findings reported in the manuscript and then comment on their novelty. I then adress a number of points that should be addresssed by the authors.

Key Findings:

A) Shortening of the solenoid and number of repeats produced an incremental and reproducible decrease in ice nucleation temperature.

The above insight are noteworthy but have been demonstrated several times using both experimental and theoretical works.

- Qiu et al JACS 2019 has e.g. a Figure entitled “Ice nucleation temperature as a function of protein length for the INP monomer of *Ps. syringae* under conditions typical of laboratory experiments. Which shows essentially the same drawn conclusion including the effect on water ordering
- Ling et al. *J. Geo. Res.* 2018 showed that by shortening the ice-nucleating site by reducing the repeats, the ice nucleation activity goes down. “The data clearly indicated that the number of repeats determines the ice nucleation temperature”
- Li et al. *Int J Biol Sci* 2012 “The expression of the full-length InaQ and of various truncated variants was induced in *Escherichia coli*”
- Several studies Warren and coworkers 1980s
-

B) The C-terminus was mutated and results were attributed to the role of these 10 basic coild in INP aggregate formation needed to efficiently nucleate ice.

The insight is interesting, but again not novel, as the experiments have been performed previously with a similar outcome. A better phrasing by the authors should be: We support the mechanism... or similarly.

- Warren et al. 1986 provided a nice review on the many reported deletions in the C-terminus.
- Ake et al. 1989 FEBS performed similar experiments and suggested that based on mutation experiments that the C-terminal unique region plays a role in functional aggregate formation.

C) Modelling of *P. borealis* using Alpha Fold 2

This information is interesting but it is worth mentioning that the identical protein structure for *P. syringae* has been solved and has been available since months <https://alphafold.ebi.ac.uk/entry/O33479> . The structure also seems to mostly confirm the models from the same authors.

D) Fluorescence imaging of the bacteria

Interesting, but largely only confirms information obtained by the same authors e.g. Vanderveer 2014 *Cryobiology*.

Major points:

- The decrease in ice nucleating activity could be explained with structural changes of the INP and a likely collapse/kink when shortening the protein. The authors must provide evidence that the proteins remain in their native fold to draw meaningful conclusions. Such experiments can be performed using CD or IR spectroscopy. With addition of bulky proteins it also seems plausible that aggregation and membrane interaction is altered which should ideally be excluded as well
- The alpha fold model described a twisted helix. The authors should comment on whether the twist has any effect on the proposed mechanisms given that AFPs are typically straight and not twisted.
- The ice nucleation data at warm temperatures raises questions and should be explained (-1 to -5). *i.* The Vali setup was reported to have uncertainty of +/- 0.2 C. Yet, the Snomax control already has a 0.6 C difference which seems like a large error. How accurate are those measurements? The statement of better resolution of Vali at higher temperature from the text seems inaccurate *ii.* The Vali setup reports activity of borealis as high as -1.1 C, while the other setups do not seem to record any freezing events at higher temperatures. That is very surprising. At least Binary should be able to detect those given that it can analyze complete freezing profiles of Snomax. (Budke et al. 2016).
- Figure 4 is unclear and the authors should report cumulative ice nucleators Nm according to Vali which is the standard in the field and which allows better comparisons across labs..
- The authors should give more credit to the existing work that spans two decades. “We agree with the hypothesis” or “we support the hypothesis” is more adequate on several occasions (e.g. abstract aggregation mechanism, INP deletion mechanism, 1% bacteria have high activity...) given that the reported results and drawn conclusions have been proposed before.

Minor points:

- The title is misleading. “potent” typically refers to the high class A/I activity in bacteria. The reported activity in this manuscript is not really high and I would suggest removing potent.
- Figure 8 is misleading as the authors reported on monomeric proteins and not the aggregation of the proteins. Moreover, different number of proteins and lengths are shown in B,C, D, making the aim and drawn conclusion from the figure ambiguous
- It would be nice to label 65 as wildtype in the Figures.
- P.3 1.68 TXT and beta-strand is occurring in more than two insect AFPs
- P.3 1.75,78 Molinero and coworkers show that water ordering is not needed (e.g. PNAS 2018) for AFPs and this should be stated in the introduction.
Alpha Fold produced a number of equally likely structure for INPs, including a kinked one. Would that have an effect on your interpretation? What is the certainty score for different AlphaFold2 runs.

Reviewer #3 (Remarks to the Author):

Apparently, no one has succeeded in crystallizing an INP so we have to rely on modeling software to predict how they interact with water molecules and nucleate ice. A 2011 study by the same group used Pymol to obtain a coiled molecule for PbINP, in which models are based on known structures with similar sequences. In the last 6 months, Alphafold has become available and, as I understand it, offers an AI- and energy-mimimization-based method to obtain more accurate structural models. The present study is an interesting demonstration of the value of this method. Using Alphafold, the authors found slight changes in the Pymol configuration that appear to enhance its ice-nucleating potential, as well as obtained predicted structures of the N- and C-terminal domains, which Pymol could not do. The authors should be congratulated on their ability to make recombinant proteins with full to zero activity depending on sequence modifications. Their analysis demonstrates the importance of long-range order for activity.

My main suggestions are related to the introduction, where more information about previous work, and rationale for the present study would be helpful.

58-84. Authors should state which INPs are described in refs. 12, 13 and 14.

What are the uncertainties and what are the unanswered questions about the Garnham model? Do the same questions apply to other INPs? Why is a new analysis needed? What can alphafold do that the other modeling programs cannot?

In what ways does PbINP resemble and differ from other INPs? This will help to show how the present results might be applicable to other INPs.

James Raymond

REVIEWER COMMENTS

Reviewer #1 (Remarks to the Author):

This manuscript by Forbes et al. reports an extensive set of experiments with synthetic ice nucleation proteins (INPs) to unravel the role of the size of the ice nucleation sites as well as the function of the N- and C-terminal structure. INPs of *Pseudomonas borealis* and mutants thereof are prepared and analysed with a whole array of ice nucleation assays. The secondary structure of the INP is predicted using the AlphaFold algorithm.

The article touches upon one of the most important and fascinating questions about INPs: What is the key driving force for ice activity? What is the role of the terminal domains? What is driving the formation of effective ice nucleating sites by assembly of INPs at the cell surface?

The authors draw a new picture of the molecular mechanisms: They identify new folding motifs in silico and relate them convincingly to experimental data. Protein engineering - truncation, blocking of site - is used as an effective experimental tool to test the hypothesis of the authors. The models are mostly supported by the data (see comments below).

This is a timely study and will be interesting to read for the large ice nucleation community but also for the broad readership of Nature Communications with a general interest science. The artwork is of high quality and the manuscript is structured well.

I recommend the article for publication after the comments below are addressed.

1. The authors predict a solenoid structure for the INP. IN the antiparallel orientation within the assemblies this means that the relative orientation of the R-coil changes with respect to the neighbouring coils. This should affect the ability to aggregate and maybe even the shape of the formed complexes. The authors should comment on this.

The hypothetical model in Figure 8B-D is designed to show how shortening the length of the central solenoid might not affect INP self-assembly but just reduce the area of the water-organizing surface, whereas loss of the R-coils could prevent self-assembly. There are equally plausible models where the INP solenoids are arranged in a parallel orientation or in a bundle rather than a sheet. The uncertainties with these models include the effect of twist in the solenoid and whether the R-coils can bind anywhere to the water-organizing coils. These possibilities could lead to branching and extension of the patch through a staggered assembly. We have added these points into the Discussion at the top of P24. *“However, the network of INPs may not adopt the arrangement depicted in Figure 8: similarly plausible models could involve INPs in a parallel orientation, or a three-dimensional bundle rather than a perfectly flat sheet. The uncertainties with these models include the effect of the twist in the solenoid and whether the R-coils can interact with any segment of the water-organizing coils. These possibilities could lead to branching and extension of the INP network through a staggered assembly. It should be noted that the AlphaFold2 model predicts less than 3 degrees of rotation between adjacent coils; this relatively small amount of twist may be eliminated by the zippering together of neighbouring solenoids to form a flat surface without unduly straining the structure. Recently, an InaZ deletion construct without its N-*

terminal domain was expressed in E. coli, leading to cytoplasmic INP retention yet the ice nucleation temperature was only 2-3 °C lower than that of full-length InaZ⁴⁷. This suggests that large INP assemblies were able to form in the cytoplasm and efficiently nucleate ice without a membrane for support. Thus, unanchored INP aggregates in solution might not produce the flat surface as depicted in Figure 8."

2. The Garnham model proposed lateral assembly into large INP patches and the formation of a joint IN surface. This is also suggested in Figure 8. How is the formation of a flat area working when the assembly consists of twisting solenoids?

The gradual twist to the central solenoid is an interesting prediction of the AlphaFold model. For a monomer in solution, it amounts to half of a rotation over the length of 65 coils, or <3 ° of rotation between one coil and the next. We speculate that this small amount of twist could be eliminated by the zipping together of neighbouring solenoids to make a flat surface without unduly straining the structure. We have added a note about this to the Discussion beginning 7 lines down on P24: *"It should be noted that the AlphaFold2 model predicts less than 3 degrees of rotation between adjacent coils; this relatively small amount of twist may be eliminated by the zipping together of neighbouring solenoids to form a flat surface without unduly straining the structure."*

3. Self assembly of the INPs in Nature takes place at a cell surface. How does this enter the model? A flat model is drawn in Figure 8 but is this a likely scenario in solution?

This is a good question, but hard to answer without the kind of structural data that a technique like cryo-electron tomography might provide. In answer to point 1 above, we have now noted in the Discussion that there are equally plausible models to the one shown in Figure 8 where the INP solenoids could be arranged in a parallel orientation or as a bundle rather than a sheet. Kassmannhuber, J. *et al.* "Freezing from the inside: Ice nucleation in Escherichia coli and Escherichia coli ghosts by inner membrane-bound ice nucleation protein InaZ." *Biointerphases* **15**, 031003 (2020), removed the export signal from an INP. When the INP was produced in the bacterial cytoplasm not attached to a membrane, these authors reported ice nucleation temperatures of just a couple of degrees lower than when the INP was exported to the outer membrane. This suggests that large INP assemblies can form without a membrane for support and still efficiently nucleate ice formation. This work was cited in the Discussion at the end of the first paragraph on P24: *"Recently, an InaZ deletion construct without its N-terminal domain was expressed in E. coli, leading to cytoplasmic INP retention yet the ice nucleation temperature was only 2-3 °C lower than that of full-length InaZ⁴⁷. This suggests that large INP assemblies were able to form in the cytoplasm and efficiently nucleate ice without a membrane for support. Thus, unanchored INP aggregates in solution might not produce the flat surface as depicted in Figure 8."*

4. Deletion of the C-terminal domain reduces ice activity. Is this explained by denaturing of the INPs?

The AlphaFold model shows how the C-terminal domain forms a solenoid capping function by covering the hydrophobic core of the solenoid while interacting with the unpaired surface of the last coil. We envisage removal of the C-terminal capping domain would destabilize several of the neighbouring R-coils that in the Figure 8 model are critical for the formation of the INP superstructure. Shifting the equilibrium from an INP patch to a monomer could explain the drastic loss of ice nucleation activity on

deleting the solenoid cap. We have covered this point in the second paragraph of the Discussion on P21: "Capping structures are essential for the stability of beta-solenoids^{15, 42}, and without them the solenoid tends to unravel or form amyloid fibrils⁴³. The complete loss of ice nucleation activity seen with the deletion of the C-terminal domain is consistent with previous findings on inaZ³⁵."

5. Deletion of domains, particularly at the termini, or addition of bulky groups can severely destabilise the folding of the modified INP relative to what was predicted by AlphaFold. This can be an additional factor for the ice activity. Ideally one would measure the secondary structure of the constructs at least with IR or CD to make sure there are no significant changes. A structure prediction with AlphaFold could also be a good way, even though I have to admit I am not sure if the effort would be reasonable. At least the possibility should be mentioned in the discussion.

The inability to date to produce INPs as stable monomers eliminates some ways of checking the structure of mutants and deletion constructs, as for example by IR and CD. This is the reason we added GFP to the C terminus of our constructs. If the INPs were misfolded and prone to degradation or inclusion body formation we should see this when viewing the *E. coli* cells by fluorescence microscopy as we routinely do for each construct and each time we grow the bacteria. What we invariably observe is an even pattern of green fluorescence throughout the cytoplasm of similar intensity for each construct. The same is true for the insertion of mRuby2 protein that added red fluorescence to the green (Figure 7). We have described these issues in the second paragraph on page 27 of the Discussion: "Insertions, deletions, and mutations of PbINP could potentially destabilize the protein fold and contribute to a decrease in activity. When we used AlphaFold to predict the structure of shortened or mutated constructs there was essentially no change in the protein structure. It would ideal if the fold of modified INPs could be assessed using IR and/or CD spectroscopy, but it has not yet been possible to isolate pure native INPs or their assemblies. Instead, their ice nucleation activity has been measured within the host bacterium in the presence of thousands of other proteins that would mask the IR or CD signals from the INPs. However, we note that none of the mutation sets caused a catastrophic loss of activity or a failure to fold the C-terminal GFP domain, which acts as an internal control for folding. This result is consistent with the mutated T and Y in the three water-organizing motifs being on the outside of the solenoid rather than contributing to the protein core, which in turn supports previous INP structural models^{12, 13}."

Reviewer #2 (Remarks to the Author):

Recommendation: This paper is not recommended for publication at Nature Communications

The manuscript submitted by Forbes *et al.* examines the working mechanism of INPs by studying the effect of modifications of the size of the ice-nucleation site of the INPs. The reported experiments were done carefully and the manuscript is well written. The topic of the manuscript is important and it contains/confirms several interesting findings. The obtained results have however largely been known/published and are not sufficiently novel to be of interest to the broad *Nature Communication* audience, and may be better suited for a more specialized journal. In addition, there a few key points that need to be addressed prior to publication in any journal.

In our opinion, this is an inadequate summary of the advances reported in Forbes *et al.* The novelty of the manuscript does not lie in the modifications to the size of the INPs, which has been explored previously as referenced in the text. But importantly, we have shown that spoiling the putative water-

organizing motifs by mutation without affecting the solenoid structure or shortening its length reduces ice nucleation activity. We are not aware of any other study reporting mutagenesis on these motifs. The significance of these findings is that our experiments make a strong mechanistic link between ice binding and ice nucleation. Another novel result is that we showed the continuity of the water-organizing surface is critical for activity. We demonstrated this by two different methods: insertion of mRuby2 into the *Pb*INP central domain, and mutagenesis within a 23-repeat segment of the repetitive region. A third novel result is our discovery that the 10 C-terminal coils (which we now call R-coils) of the central solenoid are not water-organizing but have a different role, probably that of facilitating higher-order INP structures.

I first summarize the key findings reported in the manuscript and then comment on their novelty. I then address a number of points that should be addressed by the authors.

Key Findings:

A) Shortening of the solenoid and number of repeats produced an incremental and reproducible decrease in ice nucleation temperature.

The above insights are noteworthy but have been demonstrated several times using both experimental and theoretical works.

- Qiu et al. JACS 2019 has e.g. a Figure entitled "Ice nucleation temperature as a function of protein length for the INP monomer of *Ps. syringae* under conditions typical of laboratory experiments. Which shows essentially the same drawn conclusion including the effect on water ordering

- Ling et al. J. Geo. Res. 2018 showed that by shortening the ice-nucleating site by reducing the repeats, the ice nucleation activity goes down. "The data clearly indicated that the number of repeats determines the ice nucleation temperature"

- Li et al. Int J Biol Sci 2012 "The expression of the full-length InaQ and of various truncated variants was induced in *Escherichia coli*"

- Several studies Warren and coworkers 1980s

In Forbes *et al.* we did not feel it necessary to cite every paper that has performed truncation studies but rather to acknowledge at the bottom of P7 the pioneering work of Warren and his co-workers: "Previous studies on the *P. syringae* INP gene *inaZ* (encoding 61 solenoid coils) expressed in *E. coli* have shown that deletions of 1-34 repeats from the central repetitive region reduced the ice nucleation temperature by up to 8.5 °C, in a manner generally consistent with deletion size^{27, 35}". What is different about our truncation experiments is that they are systematic and have defined the limit of an INP length below which there is no detectable ice nucleation. These experiments required the use of two super-sensitive ice nucleation devices that would not have been possible years ago prior to their development. Of the three papers listed by Reviewer #2 (above), the 2012 report by Li *et al.*, is only about truncation of the N-terminal region for surface display and is cited on P26. The 2018 paper by Ling *et al.*, is discussed and cited on P24, and the 2019 reference by Qiu *et al.* is cited on P4.

B) The C-terminus was mutated and results were attributed to the role of these 10 basic coils in INP aggregate formation needed to efficiently nucleate ice.

The insight is interesting, but again not novel, as the experiments have been performed previously with a similar outcome. A better phrasing by the authors should be: We support the mechanism... or similarly.

- Warren et al. 1986 provided a nice review on the many reported deletions in the C-terminus.

- Ake et al. 1989 FEBS performed similar experiments and suggested that based on mutation

experiments that the C-terminal unique region plays a role in functional aggregate formation.

We have expanded upon this point in the second paragraph of the Discussion (P21) and have included the Abe *et al.* citation (reference 44): “The complete loss of ice nucleation activity seen with the deletion of the P_bINP C-terminal domain is consistent with previous findings in the literature on *inaZ*³⁵ and *Erwinia ananas* INP⁴⁴. Indeed, AlphaFold predicted an almost identical structure for the C-terminal domain of *inaZ* to that of P_bINP (not shown). What we have added here is the definition and boundaries of the capping structure and the discovery of the adjacent R-coils that are equally important for ice nucleation activity.” Also, these earlier studies did not check if a C-terminal deletion caused misfolding of the entire protein. Again, all our constructs have a C-terminal GFP domain that gives some assurance that the protein is folded.

C) Modelling of *P. borealis* using Alpha Fold 2

This information is interesting but it is worth mentioning that the identical protein structure for *P. syringae* has been solved and has been available since months

<https://alphafold.ebi.ac.uk/entry/O33479>.

The structure also seems to mostly confirm the models from the same authors.

It is well known that the developers tested AlphaFold by modeling hundreds of different proteins. What we have added here is commentary and insight on the model. We show that the unknown C-terminal domain has all the attributes of a solenoid capping structure. The developers of the AlphaFold INP model did not highlight any difference between the C-terminal 10 solenoid coils and the other coils. This was shown here by our alignment studies and Web Logo plots. AlphaFold predicts a twist to the solenoid that Referee 1 has picked up on and deserves commentary, which we have provided in answer to their first point. See P24 of the revised manuscript.

D) Fluorescence imaging of the bacteria

Interesting, but largely only confirms information obtained by the same authors e.g. Vanderveer 2014 *Cryobiology*.

There appears to be a misunderstanding regarding the reason for GFP tagging. We did not simply repeat previous work here: we used their method but for a different reason, namely, to show that all our constructs are folded and uniformly produced and distributed through the *E. coli* cytoplasm.

Major points:

- The decrease in ice nucleating activity could be explained with structural changes of the INP and a likely collapse/kink when shortening the protein. The authors must provide evidence that the proteins remain in their native fold to draw meaningful conclusions. Such experiments can be performed using CD or IR spectroscopy. With addition of bulky proteins it also seems plausible that aggregation and membrane interaction is altered which should ideally be excluded as well

In the ~50 years that INP have been investigated no one has produced a soluble, monomeric INP for structural characterization. Therefore, techniques like CD and IR are, unfortunately, not applicable in whole bacteria where INP is just one of thousands of cellular proteins. However, we argue that the steady decline in ice nucleation temperatures as the solenoid is shortened (Figure 3) indicates that there is no abrupt collapse/kink in the tertiary structure of the central domain. We have seen no evidence that

GFP tagging has negatively impacted the activity of *PbINP*. Instead, it has provided assurances that the INP is properly folded.

- The alpha fold model described a twisted helix. The authors should comment on whether the twist has any effect on the proposed mechanisms given that AFPs are typically straight and not twisted.

We have commented on this in answer to Referee 1's first point and have added a note about this to the Discussion on P24: *"It should be noted that the AlphaFold2 model predicts less than 3 degrees of rotation between adjacent coils; this relatively small amount of twist may be eliminated by the zippering together of neighbouring solenoids to form a flat surface without unduly straining the structure."*

- The ice nucleation data at warm temperatures raises questions and should be explained (-1 to -5). *i.* The Vali setup was reported to have uncertainty of +/- 0.2 C. Yet, the Snomax control already has a 0.6 C difference which seems like a large error. How accurate are those measurements? The statement of better resolution of Vali at higher temperature from the text seems inaccurate *ii.* The Vali setup reports activity of borealis as high as -1.1 C, while the other setups do not seem to record any freezing events at higher temperatures. That is very surprising. At least Binary should be able to detect those given that it can analyze complete freezing profiles of Snomax. (Budke et al. 2016).

We used the Vali-type apparatus for initial screening of constructs before sending them for detailed analysis with the BINARY and WISDOM apparatuses. Since none of the ice nucleation results in the main paper were obtained on the Vali-type apparatus, we have removed mention of it from the text and have deleted the two supplementary tables S2 and S3.

- Figure 4 is unclear and the authors should report cumulative ice nucleators Nm according to Vali which is the standard in the field and which allows better comparisons across labs..
 - The authors should give more credit to the existing work that spans two decades. "We agree with the hypothesis" or "we support the hypothesis" is more adequate on several occasions (e.g. abstract aggregation mechanism, INP deletion mechanism, 1% bacteria have high activity...) given that the reported results and drawn conclusions have been proposed before.

Nm spectra show the distribution of ice-active sites at different temperatures and are often used in our community where different substances are compared, or in cases where f_{ice} curves alone were not a good representation. This is one way to normalise the basic physical distinctions of different substances, such as surface area. However, in this study, a single characteristic freezing temperature was observed on both WISDOM and BINARY, and when the sample was diluted, the temperature of heterogeneous freezing remained relatively stable. Therefore, we decided against a presentation of Nm spectra and rather show the f_{ice} representation that is probably more familiar to the biological community of readers. What this analysis shows is that the *E. coli* expressing *PbINP* have remarkably uniform activities and that the activity of a one cell is the same as that of the whole culture.

Minor points:

- The title is misleading. "potent" typically refers to the high class A/I activity in bacteria. The reported activity in this manuscript is not really high and I would suggest removing potent.

We are using 'potent' in the normal/broad sense of the word.

- Figure 8 is misleading as the authors reported on monomeric proteins and not the aggregation of the proteins. Moreover, different number of proteins and lengths are shown in B,C, D, making the aim and drawn conclusion from the figure ambiguous

At no point in the paper do we claim to be working with monomeric proteins. All our results are presumed to be from INP aggregates or oligomers. We have added text to the Discussion of Figure 8 on P23 to make this clearer: *“Modest shortening of the water-organizing coils would not stop the ability of the INPs to associate, hence the relatively minor reduction in ice nucleation temperatures for aggregates of PbINPs with partially deleted water-organizing regions explored in our study.”*

- It would be nice to label 65 as wildtype in the Figures.

The wild-type sample is *PbINP* expressed in *P. borealis*. When the 65-coil *PbINP* is produced in *E. coli* it is already modified with a C-terminal GFP and we hesitate to call that ‘wild type’.

- P.3 I.68 TXT and beta-strand is occurring in more than two insect AFPs

Here we have changed ‘two’ to ‘several’ and added another reference on P3 of the Introduction: *“.....where the two threonine residues on the beta-strand project outwards as they do in several insect AFPs^{16, 17, 18}.”*

- P.3 I.75,78 Molinero and coworkers show that water ordering is not needed (e.g. PNAS 2018) for AFPs and this should be stated in the introduction.

We beg to differ on this point. ‘Showing’ something implies an experimental demonstration with appropriate controls. We highly appreciate modeling, and we accept that it can provide a valuable guide to proving something by experimentation. We also disagree on this point because AFPs and INPs share some of the same water-ordering motifs, and our mutagenesis experiments support their involvement in the ice nucleation mechanism.

Nevertheless, we have followed the Reviewer’s suggestion to include the Hudait *et al.*, PNAS 2018 paper (reference 25) in the Introduction near the top of P4 and the statement that water ordering is not needed by AFPs: *“However, this hypothesis was not supported by vibrational sum-frequency generation spectroscopy²⁴ or by a modeling study²⁵, which suggested that water ordering is not essential for AFP activity.”* This will help frame the disagreement about the mechanism of action being based on water organization. Drawing attention to these differences of opinion with help science move forward.

Alpha Fold produced a number of equally likely structure for INPs, including a kinked one. Would that have an effect on your interpretation? What is the certainty score for different AlphaFold2 runs.

Overall, the AlphaFold models show consistency in their structures from run to run. Only one run in five showed a kink in the solenoid. Confidence was lowest in the regions that appear to be a random coil, which includes the proline-rich linker region in the N-terminal domain and the short tail at the end of the C-terminal domain. The small variation that does exist in the repetition of the models leads us to think that AlphaFold2 is not overfitting the data.

Reviewer #3 (Remarks to the Author):

Apparently, no one has succeeded in crystallizing an INP so we have to rely on modeling software to predict how they interact with water molecules and nucleate ice. A 2011 study by the same group used Pymol to obtain a coiled molecule for PbINP, in which models are based on known structures with similar sequences. In the last 6 months, Alphafold has become available and, as I understand it, offers an AI- and energy-mimimization-based method to obtain more accurate structural models. The present study is an interesting demonstration of the value of this method. Using Alphafold, the authors found slight changes in the Pymol configuration that appear to enhance its ice-nucleating potential, as well as obtained predicted structures of the N- and C-terminal domains, which Pymol could not do. The authors should be congratulated on their ability to make recombinant proteins with full to zero activity depending on sequence modifications. Their analysis demonstrates the importance of long-range order for activity.

My main suggestions are related to the introduction, where more information about previous work, and rationale for the present study would be helpful.

58-84. Authors should state which INPs are described in refs. 12, 13 and 14.

This is a good point. In the second paragraph of the Introduction, we were discussing the INPs of both *Pseudomonas syringae* and *Pseudomonas borealis*. Although these INPs are similar enough for generalizations to be made about their structures, we have now distinguished from which species the INPs were derived. Thus, we have updated the 2nd paragraph of the Introduction: “In 1993, Kajava and Lindow modeled the central domain of InaZ from *P. syringae* as a series of antiparallel β -sheet clusters in a planar zigzag arrangement that enabled interdigitation of monomers to form large aggregates¹⁴” . “The *P. syringae* INP model of Graether and Jia¹³ was built....” and: “In contrast, the model of Garnham et al.¹² was guided by the beta-roll structure common to RTX proteins but it also produced a beta-solenoid structure for a section of eight 16-residue coils from *P. borealis* INP (PbINP).”

What are the uncertainties and what are the unanswered questions about the Garnham model? Do the same questions apply to other INPs? Why is a new analysis needed? What can alphafold do that the other modeling programs cannot?

The Garnham model proposed that the conserved Tyr-Gly-Ser motif in each coil of the central solenoid dimerized to the antiparallel arrangement of the same sequence in the coils of a neighbouring INP solenoid. An unanswered question about this model was how this dimerization might lead to more extensive networks of INPs that would be large enough to trigger ice nucleation at high sub-zero temperatures. The central repetitive region of different INPs are similar enough for the Garnham model to be common to all. However, a new analysis was timely because, as mentioned in the manuscript, the 2015 discovery of an antifreeze protein from midges showed that a stacked array of Tyr could serve as an ice-binding site (reported at the bottom of P 26). Thus, in the context of INPs, stacked Tyr may help to organize more ice-like waters on the INP surface. The recent introduction of the powerful modeling program – AlphFold2 – provided the means to do the new analysis in an impartial way. The key difference in the models is shown in Figure 1, where the Ser of the Tyr-Gly-Ser motif points inside the

coils in the AlphaFold model where it cannot support dimerization (discussed in the top paragraph on P 27).

In what ways does PbINP resemble and differ from other INPs? This will help to show how the present results might be applicable to other INPs.

With reference to Figure 1 we have added a sentence second from the bottom of P 5 to describe the conservation of different regions across the INP model: *“Based on multiple sequence alignments of the INPs used in Figure S1, the C-terminal domain shows extensive sequence identity, as do their central repetitive domains despite differences in the total number of 16-residue repeats that range from ~50 to ~80.”* The portion of the alignment shown in Figure S1 is the N-terminal region and illustrates good conservation of the first half that corresponds to the globular region in the AlphaFold structure prediction (Figure 1) but poor conservation of the disordered linker region. We have now added this sentence to the end of P 5: *“The portion of the alignment shown in Figure S1 is the N-terminal region and illustrates good conservation of the first half that corresponds to the globular region in the AlphaFold structure prediction (Figure 1) but poor conservation of the disordered linker region.”*

James Raymond

REVIEWERS' COMMENTS

Reviewer #1 (Remarks to the Author):

This is an impressive manuscript. The revision is addressing all concern I had with the presentation of the work.

Reviewer #2 (Remarks to the Author):

he authors have not fully convinced me that the results are novel e.g. doi:
<https://doi.org/10.1101/2022.01.21.477219> .

Besides that, the authors addressed most of my comments sufficiently, but two points still need to be clarified.

1. The intact structural fold of the INPs has not been shown. This makes alternative explanations very plausible, e.g. that mutants at a certain size simply unfold and hence activity drops. CD spectroscopy has been shown in the past to be useful for ice nucleation data. e.g. Schmidt et al. 1997 FEBS, Hartmann et al. 2022, Roeters et al. 2021

The authors should perform CD measurements as it would significantly strengthen the conclusions.

2. Nm plots for the concentration series should be reported. The article is published in a journal targeting a broad audience including the atmospheric community. Nm plots are also standard for dilution series in the biological community as it allows better comparisons to other studies. Figure 4 should show Nm plots!

Reviewer #2 (Remarks to the Author):

The authors have not fully convinced me that the results are novel e.g. doi:

<https://doi.org/10.1101/2022.01.21.477219> .

This informative manuscript by Hartmann *et al.* was posted after we submitted our paper to Nature Communications. Much of what is novel about our paper (the decrease in ice nucleation activity on mutating putative water-organizing motifs, the importance of a continuous water-organizing surface, the discovery of R-coils) is not covered in the Hartmann manuscript and remains novel.

Besides that, the authors addressed most of my comments sufficiently, but two points still need to be clarified.

1. The intact structural fold of the INPs has not been shown. This makes alternative explanations very plausible, e.g. that mutants at a certain size simply unfold and hence activity drops. CD spectroscopy has been shown in the past to be useful for ice nucleation data. e.g. Schmidt et al. 1997 FEBS, Hartmann et al. 2022, Roeters et al. 2021

The authors should perform CD measurements as it would significantly strengthen the conclusions.

1. CD was not an option for our studies because all our ice nucleation assays were done on whole bacteria. The fluorescence signal we see from the GFP-tagged *Pb*INP is uniform across the cytoplasm in all our INP constructs gives an assurance that the protein is soluble and not aggregated in inclusion bodies due to misfolding.

2. Nm plots for the concentration series should be reported. The article is published in a journal targeting a broad audience including the atmospheric community. Nm plots are also standard for dilution series in the biological community as it allows better comparisons to other studies. Figure 4 should show Nm plots!

2. As requested by Referee #2 we have included an Nm plot of the data shown in Fig. 4 as a figure in Supplementary Information. The conclusions from the two plots are the same.